# The tail domain of the plant kinesin-12 POK2 is a versatile interaction hub

Shu Yao Leong[1], Laura Muras[1], Benedikt S. J. Fischer[1], Sehee Jang[1], Anastasia Gurskaya[1], Mayank Chugh[2,3], Serapion Pyrpassopoulos[1], Hauke Drechsler[1] and Erik Schäffer[1,*]

## ABSTRACT

Tissue development and function rely on correct cell patterning. In plants, patterns are determined by the new cell wall formed during mitosis. During preprophase in embryophytes, ring-like positional cues already mark the future division plane on the plasma membrane. These cues include the kinesin-12 motors PHRAGMOPLAST ORIENTING KINESIN 2 (POK2, also known as KIN12D) and its paralogue POK1 (KIN12C). They are essential for correctly aligning the phragmoplast – a microtubule scaffold for cell wall formation. Although we have a basic understanding of how these motors align the phragmoplast, we currently lack information on how they are targeted to and maintained at the plasma membrane. Here, we reconstituted recombinant POK2 tail fragments on microtubules and vesicles *in vitro*. We found that the POK2 tail interacted with microtubules, the microtubule-associated protein MAP65-3, as well as with certain anionic lipids. We identified a short element in the POK2 tail responsible for all three interactions. Our data suggest a sequential and cooperative mechanism that targets POK2 specifically to the future division site. There, it is robustly maintained through interactions with the plasma membrane to establish the cell division plane.

KEY WORDS: Plant cell division, Phragmoplast orienting kinesins, Kinesin-12, Microtubules, *In vitro* reconstitution

## INTRODUCTION

Precise placement of the cell division plane is essential for cell patterning in plant tissues. Division plane positioning begins in the S-G2 phase with the migration of the nucleus to the future division plane (Facette et al., 2019; Wada, 2018). In preprophase of embryophytes, cortical microtubules coalesce into a cortical ring around the nucleus called the preprophase band (PPB) (Pickett-Heaps and Northcote, 1966), which demarcates the future division plane. The PPB recruits molecular motors like the functionally redundant *Arabidopsis thaliana* PHRAGMOPLAST ORIENTING

[1]Center for Plant Molecular Biology (ZMBP), University of Tübingen, 72076 Tübingen, Germany. [2]Department of Systems Biology, Harvard Medical School, Boston, MA 02115, USA. [3]Biology Department, The College of William and Mary, Williamsburg, VA 23187-8795, USA.

*Author for correspondence (erik.schaeffer@zmbp.uni-tuebingen.de)

S.Y.L., 0000-0002-0312-5664; L.M., 0009-0002-8239-1385; B.S.J.F., 0009-0008-5713-2887; S.J., 0009-0007-7020-5668; A.G., 0009-0008-5281-4889; M.C., 0000-0002-5167-6890; S.P., 0000-0002-9728-9239; H.D., 0000-0001-5075-0027; E.S., 0000-0001-7876-085X

KINESIN 1 and 2 (POK1 and POK2, also known as KIN12C and KIN12D, respectively; Lipka et al., 2014; Chugh et al., 2018) as well as microtubule-associated proteins (MAPs) such as MAP65-4 (Van Damme et al., 2004; Smertenko et al., 2008; Li et al., 2017), IQ67 DOMAIN proteins 6, 7 and 8 (IQD6, IQD7 and IQD8, respectively; Kumari et al., 2021), and TANGLED (Smith et al., 2001; Walker et al., 2007). These proteins remain at the cell cortex when the PPB disassembles after the onset of mitosis and continue to mark this area as the cortical division zone (CDZ) but vary in the width of CDZ that they occupy (Van Damme et al., 2011; Smertenko et al., 2017). The CDZ serves as a guidepost for the alignment of the phragmoplast – a bipartite array of parallel microtubules. The phragmoplast is repurposed from the anaphase spindle and scaffolds the formation of the cell plate – the new cell membrane and wall – at the phragmoplast midzone where the microtubule plus-ends of the two arrays meet (Otegui et al., 2005; Smertenko et al., 2017). Vesicles with building materials are transported along the microtubules to the midzone where they fuse to form the growing cell plate. The phragmoplast expands radially by polymerising new microtubules on its outer edge and depolymerising microtubules in its centre. Eventually, the cell plate merges with the maternal cell membrane at the CDZ to separate two daughter cells. The POK1 and POK2 motors are essential for correct phragmoplast alignment, with POK2 also being necessary for its timely expansion (Herrmann et al., 2018; Lipka et al., 2014; Müller et al., 2006). Accordingly, two functionally different pools of POK2 have been described to be active during plant mitosis.

From prophase onwards, a first subpopulation of POK2 accumulates at the PPB and persists at the CDZ throughout mitosis (Herrmann et al., 2018). Truncation experiments *in vivo* suggest that an as-yet-unspecified C-terminal domain in the tail of POK2, comprising amino acids 2083–2771, is essential to its recruitment and maintenance at the CDZ. The motor activity of the POK2 N-terminal region (amino acids 1–589) only contributes to the targeting efficiency during the initial recruitment but not to the cortical association of POK2 itself (Herrmann et al., 2018). POK2 motor activity is crucial for phragmoplast orientation, as POK2$^{2083–2771}$ lacking the motor domain fails to align the phragmoplast with the CDZ in a *pok1 pok2* mutant background, despite being properly recruited to the CDZ (Herrmann et al., 2018). Our previous *in vitro* characterisation of the C-terminally truncated, minimal motor POK2$^{1–589}$ showed that POK2 is a weak, microtubule plus-end-directed motor that can switch between directed and diffusive modes (Chugh et al., 2018). We proposed that cortical POK2 gathers and acts on the plus-ends of peripheral phragmoplast microtubules, tethering them to the plasma membrane. Plus-end-directed POK2 motor activity applies pushing forces on the peripheral microtubules and the phragmoplast poles, aligning the overall structure perpendicular to the cell division plane (Chugh

et al., 2018). At the same time, the cortical POK2 motors experience counter forces that might contribute to the ongoing focussing of the CDZ into the narrower cell plate fusion site in late cytokinesis (Gunning and Wick, 1985; Lipka et al., 2014; Rasmussen et al., 2011; Walker et al., 2007; Chugh et al., 2018).

With the onset of cytokinesis, a second subpopulation of POK2 accumulates at the phragmoplast midzone and remains there during radial phragmoplast expansion until it merges with the POK2 subpopulation at the CDZ (Herrmann et al., 2018). Mutational analyses suggest that POK2 accumulates due to its plus-end-directed motor activity at the phragmoplast midzone, where it is then actively retained by the midzone-resident protein MAP65-3 (Herrmann et al., 2018). Retention might be mediated by a direct interaction between MAP65-3 and the unstructured N-terminal extension (amino acids 1–189) of POK2 or an as-yet-uncharacterised C-terminal domain in the POK2 tail (Herrmann et al., 2018). Alternatively, as in vivo data suggest that the POK2 tail might harbour a second ATP-independent microtubule-binding site (Herrmann et al., 2018), midzone retention may be due to a higher affinity of POK2 to antiparallel microtubule overlaps created by crosslinking MAP65-3 dimers. Although it is unknown how POK2 operates mechanistically at the phragmoplast midzone, the results of deletion experiments suggest that both POK1 and POK2 contribute to the proper alignment of microtubule plus-ends and the phragmoplast midzone within the cell division plane, which is required for timely expansion of the phragmoplast (Herrmann et al., 2018; Lipka et al., 2014).

Although there are hypotheses on how the two POK2 subpopulations at the CDZ and at the phragmoplast midzone might function, we do not know how plant cells regulate these POK2 subpopulations spatiotemporally. To address this question, we first need to understand how POK2 is recruited to and retained at the CDZ by identifying the binding partner for the POK2 tail at the PPB and CDZ. The PPB microtubules are an obvious candidate based on the putative second microtubule-binding site in the POK2 tail (Herrmann et al., 2018). However, this hypothesis is challenged by experiments showing that the pool of POK2 associated with the PBB and CDZ persists at the cortex upon induced microtubule depolymerisation (Herrmann et al., 2018). MAP65-3 is another potential targeting factor for POK2, as it binds to the POK2 tail and localises to the PPB. However, MAP65-3 disappears from the cortex early in the cell cycle upon PPB disassembly and, therefore, cannot be responsible for the long-term retention of POK2 at the CDZ (Caillaud et al., 2008). Furthermore, the cortical localisation of POK2 is not altered in MAP65-3-deficient cells (Herrmann et al., 2018). Although microtubules and MAP65-3 might contribute to an efficient cortical targeting of POK2, currently available data strongly suggest that there must be another, as-yet-unknown interacting element at the cortex.

To identify this element and to dissect the contribution of microtubules and MAP65-3 to CDZ targeting of POK2, we expressed the C-terminal part of the POK2 tail in insect cells, purified the recombinant protein and reconstituted it with the respective interactor candidates in vitro. We show that the POK2 tail binds directly to and diffuses along microtubules, confirming the hypothesis of a second microtubule-binding site. Furthermore, MAP65-3 interacts with the POK2 tail, increasing its microtubule-binding activity. Finally, we provide evidence that the POK2 tail directly binds to anionic lipids present in the cell membrane. We hypothesise that recruitment of POK2 to the PPB and CDZ occurs by sequential and cooperative interactions with the cortical microtubules, MAP65-3 and the cortical membrane at the CDZ.

## RESULTS

### The C-terminal domain of POK2 directly binds microtubules via their E-hooks

The initial recruitment of full-length POK2 to the PPB and CDZ is microtubule dependent but does not require its motor activity, because a POK2 tail fragment lacking the motor domain (amino acids 2083–2771) still localises correctly in vivo (Herrmann et al., 2018). This finding suggests that the POK2 tail either binds microtubules directly or is recruited to the PPB microtubules by another MAP. To test whether the POK2 tail binds microtubules directly, we expressed a recombinant, eGFP-tagged C-terminal fragment of POK2, eGFP–POK2$^{2083–2771}$ (symbolised by a blue bar throughout the figures), in insect cells and purified it by immobilised metal affinity chromatography (Fig. 1A; Fig. S1A). We incubated this fragment with surface-immobilised, guanosine-5'-[(α,β)-methyleno]triphosphate (GMPCPP)- and paclitaxel-stabilised (unless stated otherwise), rhodamine-labelled microtubules. To visualise microtubules and proteins, we used total internal reflection fluorescence (TIRF) microscopy (Schellhaus et al., 2017; Simmert et al., 2018). We found that eGFP–POK2$^{2083–2771}$ bound microtubules in a concentration-dependent, cooperative manner (Fig. 1B), quantitatively described by a Hill–Langmuir equation with a dissociation constant $K_D$=106± 4 nM and a Hill coefficient $n_H$=4.6±0.8 (mean±fit error; Fig. 1C). The large Hill coefficient reflects cooperative binding of this POK2 fragment to microtubules, in contrast to non-cooperative Michaelis–Menten-like binding (dashed line in Fig. 1C). Cooperative binding is consistent with eGFP–POK2$^{2083–2771}$ mainly forming dimers and occasionally tetramers in solution, as judged by the running behaviour of native complexes during blue native polyacrylamide gel electrophoresis (PAGE) (Fig. S1C). Notably, eGFP–POK2$^{2083–2771}$ displayed a strong propensity to bind microtubules in bright clusters (Fig. 1B), whose intensities exceeded those of single dimeric or tetrameric particles (compare with Fig. 3A). As these clusters formed only at higher concentrations in situ on microtubules, did not contribute to background binding next to microtubules and could be dissolved by dilution of the protein, we attribute cluster formation to specific protein–protein interactions rather than to unspecific protein aggregation. Additionally, these clusters might not correspond to the cortical puncta formed by the same construct upon overexpression in interphase cells in vivo (Herrmann et al., 2018). In vitro eGFP–POK2$^{2083–2771}$ clusters form reversibly in a microtubule-dependent manner, whereas interphase puncta in vivo do not localise to microtubules, suggesting either that clusters and puncta have different origins or that additional factors must be involved during puncta formation in vivo.

Given the size of this ∼700-amino-acid C-terminal domain, we next sought to narrow down the microtubule-binding domain by further truncating the tail domain. Based on in silico predictions highlighting coiled-coil (DeepCoil2; https://toolkit.tuebingen.mpg.de/tools/deepcoil2; Zimmermann et al., 2018) and disordered regions (AIUpred; https://aiupred.elte.hu/; Erdős and Dosztányi, 2024) (Fig. 1A), we then cloned, expressed and purified three additional truncated versions of the POK2 C-terminal domain, with eGFP–POK2$^{2083–2440}$ covering the first half (represented by a light green bar in the figures), eGFP–POK2$^{2510–2771}$ covering the second half (represented by an orange bar in the figures) and eGFP–POK2$^{2510–2677}$, which corresponds to the second half lacking the very C-terminal end (represented by a pink bar in the figures; Fig. 1A; Fig. S1B). When tested for their microtubule-binding capacities, as described above, only eGFP–POK2$^{2510–2771}$ retained significant microtubule-binding activity.

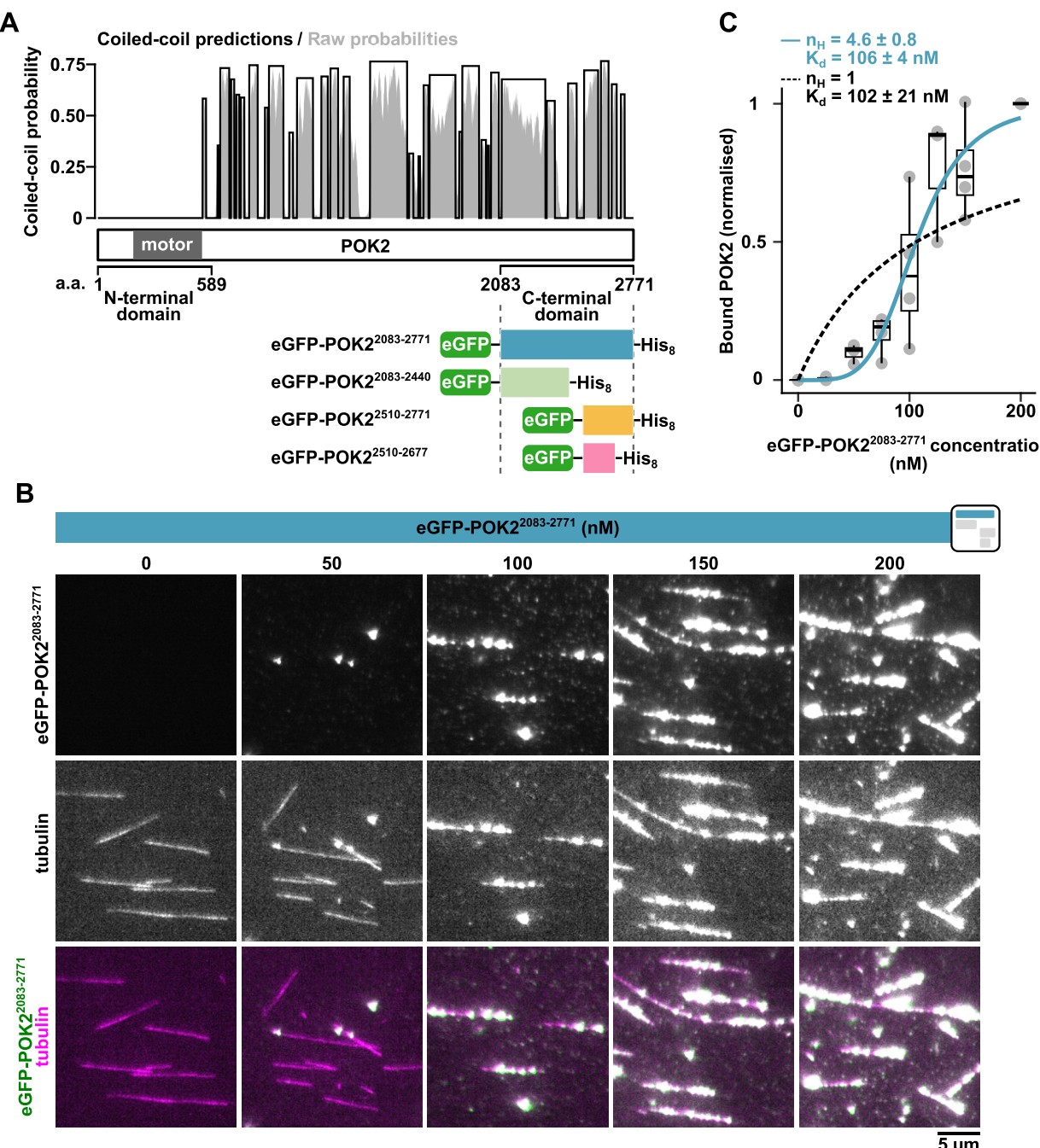

**Fig. 1. The C-terminal domain of POK2 binds microtubules in the mid-nanomolar range.** (A) Coiled-coil prediction by DeepCoil2. Scheme below depicts the C-terminal fragments of POK2 that have been designed for this study based on the predicted coiled-coil patterning (a.a., amino acids). (B) TIRF microscopy images of recombinant eGFP–POK2$^{2083–2771}$ at increasing concentrations binding to GMPCPP- and paclitaxel-stabilised rhodamine-labelled microtubules *in vitro*. There was bleed-through of the POK2 signal into the tubulin channel (middle row). The icon on the top right indicates the construct used, based on Fig. 1A. (C) Binding curve of eGFP–POK2$^{2083–2771}$. Plot shows the relative amount of bound eGFP–POK2$^{2083–2771}$ (i.e. intensity per microtubule length) at a given concentration normalised to the saturating condition at 200 nM. Data points are mean of four repeats. Boxes and lines mark the interquartilie range and the median, respectively; whiskers mark the Tukey range. Dashed black line, Michaelis–Menten fit; blue solid line, Hill fit.

However, the affinity was at least an order of magnitude smaller compared to that of the largest construct, eGFP–POK2$^{2083–2771}$ (Fig. 2A,B, compare to Fig. 1B,C). Deletion of amino acids 2083–2509 suppressed multimerisation of the POK2 tail in solution (i.e. for constructs eGFP–POK2$^{2510–2771}$ and eGFP–POK2$^{2510–2677}$; Fig. S1C), suggesting that multimerisation and the concomitant generation of multivalency is required to increase the affinity of

the POK2 tail to the nanomolar range. Constructs lacking the very C-terminal end (i.e. amino acids 2678–2771), however, only bound microtubules poorly (Fig. 2A,B) irrespective of their multimerisation state in solution (dimeric eGFP–POK2$^{2083–2440}$ and monomeric eGFP–POK2$^{2510–2677}$; see Fig. S1C).

In summary, these data suggest that the final ~100 amino acids of POK2 are critical for microtubule binding. An unusual

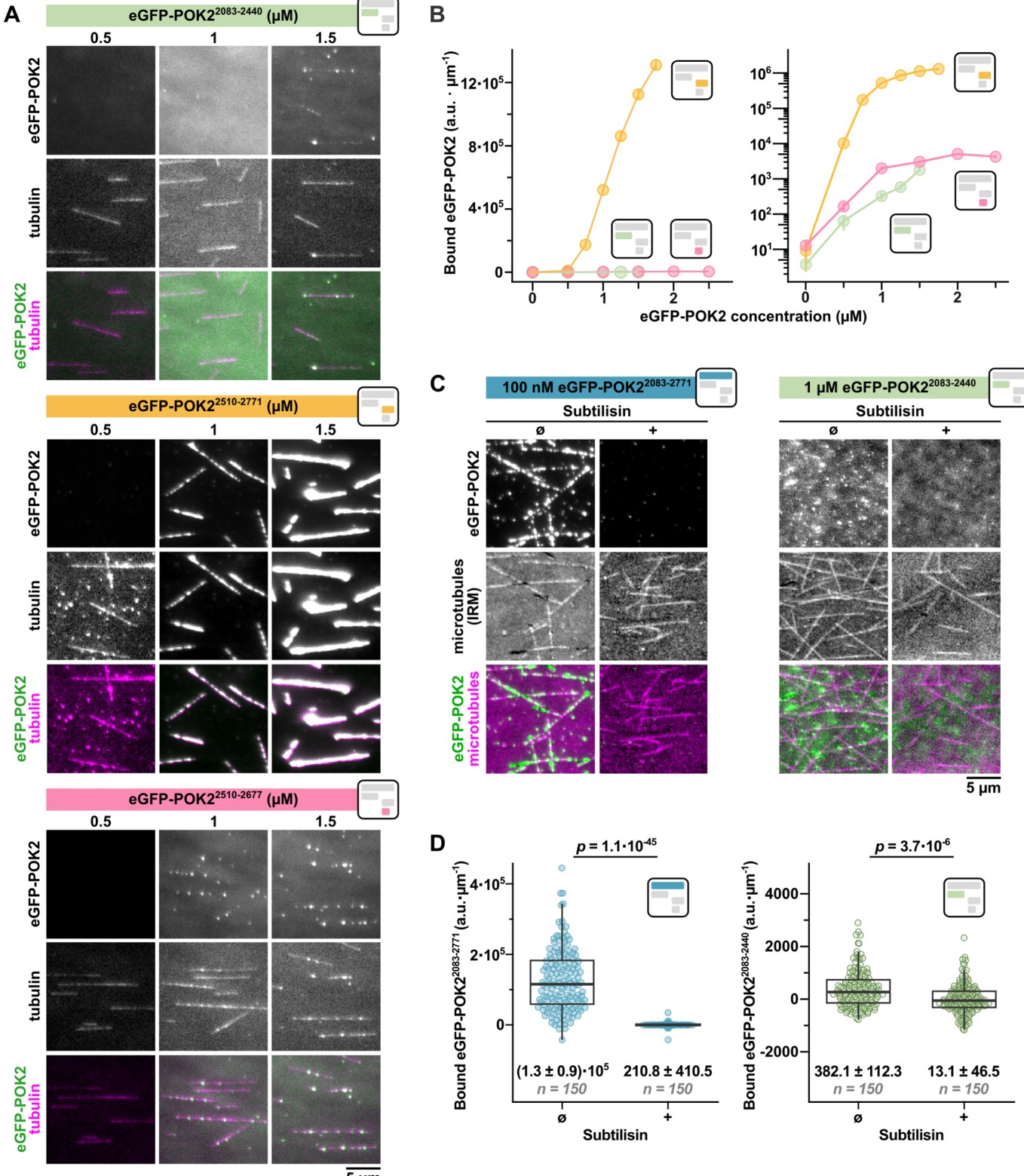

**Fig. 2.** See next page for legend.

accumulation of amino acids with positively charged side chains within the last ~50 amino acids of POK2 (Fig. S1E) prompted us to test whether the POK2 tail electrostatically binds to the negatively charged C terminus of tubulin, called the E-hook (Kikkawa et al., 2000). As hypothesised, eGFP–POK2$^{2083–2771}$

did not bind stabilised microtubules that had their E-hooks removed using the subtilisin protease (Fig. S1D), indicating that binding depends on electrostatic interactions (Fig. 2C,D). The poor microtubule-binding activity of eGFP–POK2$^{2083–2440}$ was sensitive to subtilisin treatment as well (Fig. 2C,D), suggesting that there

**Fig. 2. The final stretch of POK2 containing amino acids 2678–2771 binds to the E-hooks of tubulin.** (A) TIRF microscopy images showing the binding of the indicated recombinant POK2 fragments (top rows and green in bottom rows) at the given concentrations to stabilised rhodamine-labelled microtubules (middle rows and magenta in bottom rows). Note that there is bleed-through of the POK2 signal into the tubulin channel for POK2$^{2510–2771}$ (middle and bottom rows). Brightness and contrast are uniform for each POK2 construct across different concentrations, except for the POK2$^{2510–2771}$ construct, where the magenta tubulin channels for each concentration are individually adjusted for better visibility. (B) Plots showing the mean microtubule binding of the indicated POK2 fragments (i.e. intensity per microtubule length) with a linear scale (left) and logarithmic scale (right) on the y-axis at given concentrations. Data points and vertical bars show mean ±s.e.m. of two or three repeats. Connecting lines are drawn point-to-point as a guide to distinguish data from different POK2 fragments on the logarithmic scale and are not fits to the data. (C) Microscopy images showing binding of the indicated POK2 fragments to untreated (Ø) and subtilisin-treated (+) biotinylated stabilised unlabelled microtubules. The POK2 fragments were imaged by TIRF microscopy and the microtubules were imaged by IRM. (D) Plots showing microtubule binding of the indicated POK2 fragments to untreated (Ø) and subtilisin-treated (+) biotinylated microtubules as shown in C, measured as intensity per microtubule length. Each point is a measurement from one microtubule, where data is pooled from three experimental repeats. Horizontal lines and boxes mark the median and interquartile range, respectively; whiskers mark the Tukey range. P-values are calculated from a Mann–Whitney–Wilcoxon test. Numbers indicate mean±s.e.m. Square icons depict the colour schemes and arrangement of POK2 recombinant proteins as in Fig. 1A, and are placed here for easy reference. a.u, arbitrary units.

might be additional microtubule binding N-terminally to the last ∼100 amino acids. Alternatively, the residual microtubule binding of eGFP–POK2$^{2083–2440}$ and eGFP–POK2$^{2510–2677}$ constructs might be mediated by the terminal His-tag, which in general can mediate microtubule binding in the micromolar range when fused to multimeric proteins (Inaba et al., 2022). However, their contribution to microtubule binding is at least three orders of magnitude lower than that of the ∼100 amino acids of POK2 and is therefore negligible.

**The C-terminal domain of POK2 diffuses on microtubules**
To determine how the POK2 tail interacts with microtubules, we recorded single eGFP–POK2$^{2083–2771}$ particles with TIRF microscopy as a function of time on unlabelled, stabilised microtubules that we imaged using interference reflection microscopy (IRM) (Simmert et al., 2018) (Fig. 3A). At 2 nM, the landing rate of the eGFP–POK2$^{2083–2771}$ particles was 0.16±0.01 μm$^{-1}$ min$^{-1}$ (mean±s.e.m., n=5 replicates; Fig. 3B). Once landed, 68% of the eGFP–POK2$^{2083–2771}$ particles diffused on the microtubule lattice, whereas 28% remained stationary (n=184 total number of particles). Based on the intensity distribution of the observed particles and their bleaching behaviour (Fig. 3C,D), we conclude that the observed particles were either dimers (one or two GFPs) or tetramers (as indicated by at least three GFPs; see Fig. S2), in line with our blue native PAGE experiments. Odd GFP counts could be attributed to random pre-bleaching events in solution before imaging and incomplete maturation of fluorescent proteins such that they cannot emit light (Coffman and Wu, 2012). Diffusing or stationary particles did not differ in their multimerisation state (Fig. S2), and single particles could occasionally be observed to switch between diffusive and stationary phases (4% of the events; Fig. 3A, arrowhead), arguing that both binding modes do not reflect different subpopulations of the protein. We note, however, that the diffraction-limited resolution in our TIRF assay, with single molecules having a diameter of ∼500 nm (105 nm per pixel), does not allow us to reliably discriminate between eGFP–POK2$^{2083–2771}$

particles binding statically to the microtubule (∼25 nm in width) or non-specifically to the glass surface beside the microtubule. Given that the interaction time distribution of static particles measured on the microtubule was indistinguishable from that of the background control next to the microtubule (Fig. S3A,B), we attribute most, if not all, static binding events to non-specific background binding of the protein rather than actual microtubule association. Therefore, we excluded all static events from further analyses. Such events only served as internal controls for the calculation of the diffusion coefficient and the tracking precision. The rare switching events between static and diffusive phases could also correspond to eGFP–POK2$^{2083–2771}$ particles that switch between microtubule and background binding.

Since eGFP–POK2$^{2083–2771}$ had a broad distribution of interaction times (Fig. 3A), we imaged the particles for different durations at various time resolutions to reliably describe their dynamics on microtubules. By fitting each of the respective survival plots with the sum of two exponential functions (Fig. 3E), we identified three different subpopulations that interact with the microtubules for roughly 0.9 s, 6 s and 43 s (Table S3). We do not think that bleaching, falsely interpreted as protein dissociation, reduced these time constants because eGFP–POK2$^{2083–2771}$ bleached on average after ∼670 frames, whereas each movie comprised 600 frames (Fig. 3A). To obtain a diffusion coefficient for eGFP–POK2$^{2083–2771}$, displacement histograms for lag times τ=0.1–2.0 s were plotted and multiple Gaussians were fitted to the data (Fig. 3F; Fig. S3C,D). The distributions were best described by two Gaussians reflecting the diffusive and stationary states of eGFP–POK2$^{2083–2771}$. Based on a mean squared displacement analysis, the diffusion coefficient was 0.031±0.001 μm$^2$ s$^{-1}$ (mean±fit error, n=121 traces; Fig. 3G).

**MAP65-3 increases the microtubule binding affinity of the POK2 C-terminal domain**
Although our data show that the POK2 tail can bind microtubules on its own, it does not explain how it is specifically recruited to and later maintained at the CDZ. The CDZ-resident MAP MAP65-3 is a prime candidate for early POK2 targeting, as it has been suggested to target the POK2 tail to microtubules in vivo (Herrmann et al., 2018). We hence investigated the effects of MAP65-3 on the POK2 tail–microtubule interaction. For these experiments, we expressed and purified recombinant MAP65-3–mCherry from insect cells (Fig. S4A), reconstituted it with eGFP–POK2$^{2083–2771}$ and imaged both by TIRF microscopy on unlabelled, stabilised microtubules visualised using IRM (Simmert et al., 2018).

First, we incubated 20 nM eGFP–POK2$^{2083–2771}$ with increasing amounts of MAP65-3–mCherry (0.1–20 nM). The amount of MAP65-3–mCherry bound to microtubules increased with concentration. Interestingly, the amount of microtubule-bound eGFP–POK2$^{2083–2771}$ increased concomitantly, suggesting that MAP65-3 promotes microtubule binding of this POK2 fragment (Fig. 4A–C). Microtubule binding of MAP65-3 – like that of the POK2 tail – was sensitive to subtilisin treatment. Thus, this interaction is also mediated by electrostatic interactions with the tubulin E-hooks (Fig. S5A,B; Chauhan et al., 2024). Still, cooperative binding of MAP65-3 and POK2 persisted over a range of ionic strengths (Fig. 4D,E), suggesting that both proteins form a stable heterocomplex with much higher microtubule affinity than single components.

To test which part of the C-terminal POK2 tail interacts with MAP65-3, we quantified the microtubule binding of our shorter POK2 fragments at 20 nM in the presence or absence of 20 nM

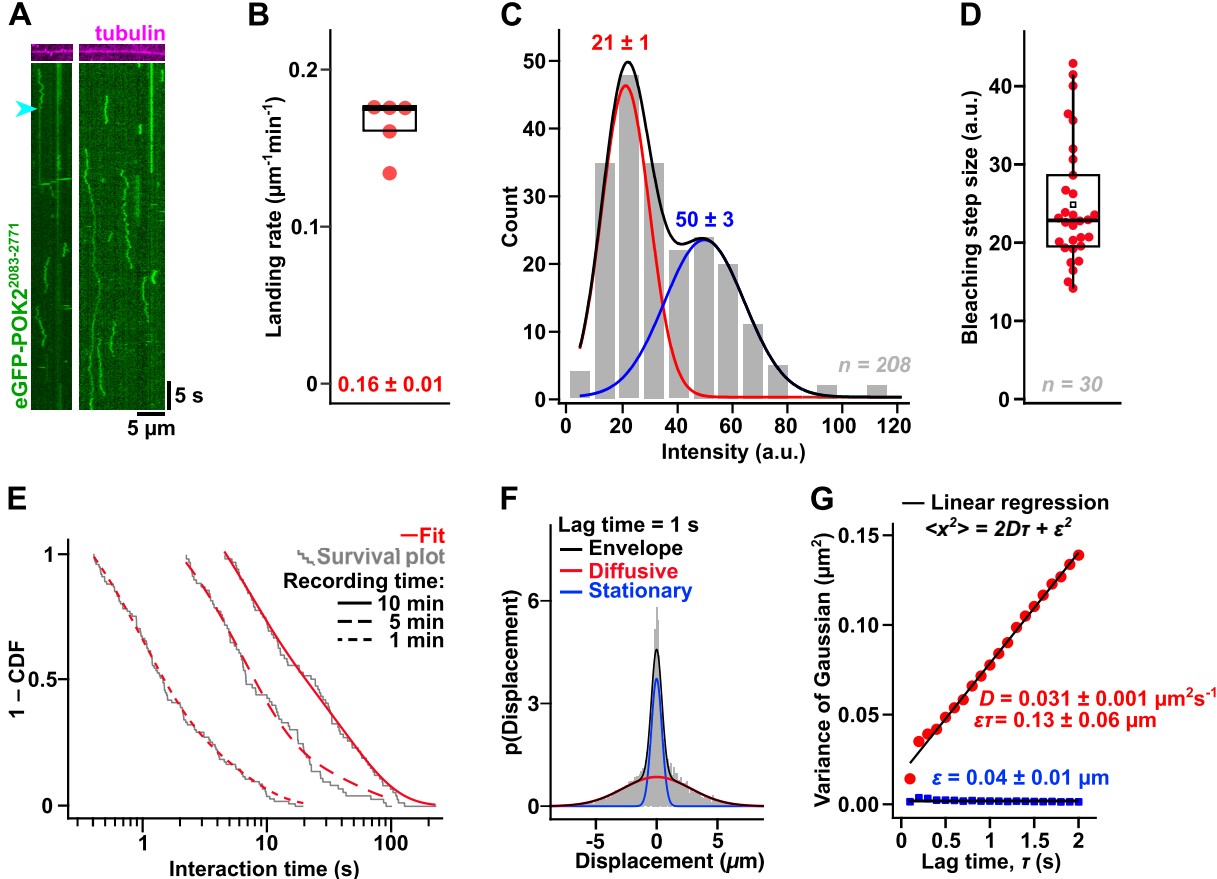

**Fig. 3. The C-terminal domain of POK2 diffuses on microtubules.** (A) Kymographs showing the movement of single eGFP–POK2$^{2083–2771}$ particles (2 nM; green, imaged with TIRF microscopy) on unlabelled stabilised microtubules (magenta, imaged using IRM). Single eGFP–POK2$^{2083–2771}$ particles can switch between diffusive and stationary binding modes (cyan arrowhead). (B) Landing rate of diffusive eGFP–POK2$^{2083–2771}$ particles on stabilised microtubules. Each data point represents the mean of five repeats (from two different flow channels). Box is the interquartile range, and the line in the box is the median (in this case, the median coincides almost with the top of the box). The mean±s.e.m. value is shown at the bottom. (C) Intensity distribution of microtubule-bound eGFP–POK2$^{2083–2771}$ particles (stationary and diffusive) in arbitrary units (a.u.). Red and blue lines are Gaussian fits to the data. Black line is the cumulative fit thereof. Peak intensity values±s.e.m. are shown. (D) Fluorescence loss during photobleaching of eGFP–POK2$^{2083–2771}$ particles in arbitrary units. Horizontal line indicates the median. Square denotes the mean. Box is the interquartile range, with whiskers being the Tukey range. Compare with B, see also Fig. S2. (E) Survival plots of the interaction times of diffusive eGFP–POK2$^{2083–2771}$ particles on microtubules taken for three recording times (1, 5 and 10 min; grey lines) sampled at 0.1, 0.5 and 1.0 s time intervals, respectively. Survival plots are fitted to sum of two exponential functions (red lines) with two time constants. See Table S3 for details. CDF, cumulative distribution function. (F,G) Single eGFP–POK2$^{2083–2771}$ particles were tracked, and distributions of the displacement steps for lag times $\tau$ (0.1 s to 2.0 s every 0.1 s) were fitted with multi-Gaussians (see Fig. S3D,E). Panel F shows a representative distribution at $\tau$=1 s. The variance of the individual Gaussians as in F was plotted against $\tau$ to generate the mean squared displacement plot in G. The diffusion coefficient, $D$, and tracking precision, $\varepsilon$, are given ±fit error, and are derived from the linear regression (black lines, see Materials and Methods for details). For the stationary population (blue), the fit was performed with $D$ set to 0. In E, $n$=103, 73, 74 traces for d$t$=0.1, 0.5, 1 s, respectively, from one or two experiments. For F–G, $n$=121 traces, pooled from two experiments.

MAP65-3–mCherry (Fig. 5A–D). Although MAP65-3–mCherry efficiently decorated microtubules (Fig. 5A), only eGFP–POK2$^{2510–2771}$ showed significant concomitant binding (Fig. 5C). Given that 20 nM eGFP–POK2$^{2510–2771}$ did not significantly bind to microtubules in the absence of MAP65-3–mCherry (see Figs 2 and 4A), we conclude that increased recruitment in the presence of MAP65-3 likely occurs by a direct interaction between both proteins. This interaction is mediated by the same C-terminal region, the last ~100 amino acids of POK2 (amino acids 2678–2771), that mediates microtubule binding.

Alternatively, binding of MAP65-3 could change the physical properties of the microtubule lattice, thereby creating a high-affinity template for the C-terminal POK2 fragment without the need for a direct interaction between both proteins. To determine whether the proteins interact directly to form a heteromeric complexes, we

observe the C-terminal POK2 fragment and MAP65-3 at the single-molecule level.

To this end, we incubated 1 nM eGFP–POK2$^{2083–2771}$ with 0.5 nM MAP65-3–mCherry on unlabelled, stabilised microtubules and imaged them using TIRF microscopy and IRM, respectively. As the single proteins and possible heterocomplexes might have different ionic strength optima for their interactions with microtubules, we tested an ionic strength range of 70–190 mM. Whereas the landing rate of eGFP–POK2$^{2083–2771}$ and MAP65-3–mCherry decreased with increasing ionic strength (due to their electrostatic nature), co-diffusing particles were most abundant at an ionic strength of 160 mM (Fig. 6A,B). Heterocomplexes either pre-formed in solution (24/44 events), or one binding partner was directly recruited from solution to a pre-bound counterpart (MAP65-3 pre-bound, 13/44 events; POK2 pre-bound, 7/44 events)

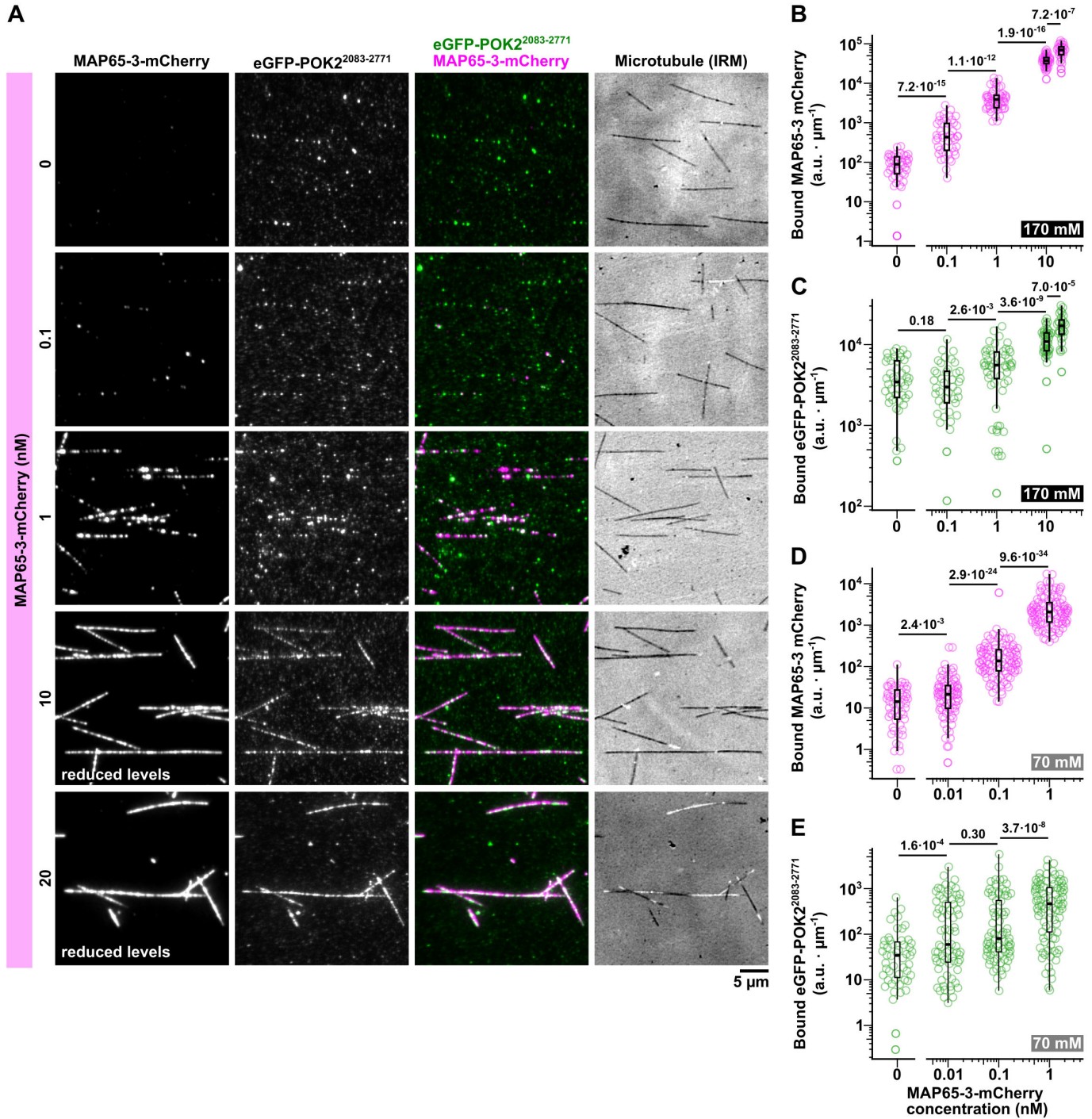

**Fig. 4. MAP65-3 increases the microtubule binding of the C-terminal domain of POK2.** (A) TIRF microscopy and IRM images showing the binding of 20 nM eGFP–POK2$^{2083–2771}$ (second column and green in third column) to microtubules (fourth column, black/white) in the presence of the indicated concentrations of MAP65-3–mCherry (first column and magenta in third column) at an ionic strength ($I$) of 170 mM. (B–D) Box plots showing the binding of MAP65-3–mCherry (B,D) and 20 nM eGFP–POK2$^{2083–2771}$ (C,E) to microtubules with increasing MAP65-3–mCherry concentrations at $I$=170 mM (B,C) and $I$=70 mM (D,E) in arbitrary intensity units (a.u.) per micrometre. Please note that the data are plotted with a logarithmic scale on the $y$-axis and that control signals for MAP65-3 at 0 nM (B,D) correspond to camera noise and background signals. Boxes are interquartile ranges, whiskers are minima and maxima, horizontal lines indicate the median, numbers indicate $P$-values calculated from a Mann–Whitney–Wilcoxon $U$-test. For B and C, $n$=45, 88, 104, 82 microtubules for 0, 0.1, 1, 10 nM MAP65–3–mCherry, respectively, from one representative of two experiments. For C and D, $n$=198, 120, 120, 140 microtubules for 0, 0.01, 1, 10 nM MAP65–3–mCherry, respectively, pooled from three experiments.

(Fig. 6C). Complexes did not form by the encounter of two different molecules on the microtubule by a search-and-capture mechanism. Based on the fluorescence intensities, the stoichiometry of POK2 and MAP65-3 was highly variable

(Fig. 6D). Such a variation is expected for heterocomplexes of proteins that can form dimers and tetramers themselves (Marciano et al., 2022). Similar to eGFP–POK2$^{2083–2771}$ by itself, heterocomplexes diffused on microtubules or were stationary

Journal of Cell Science

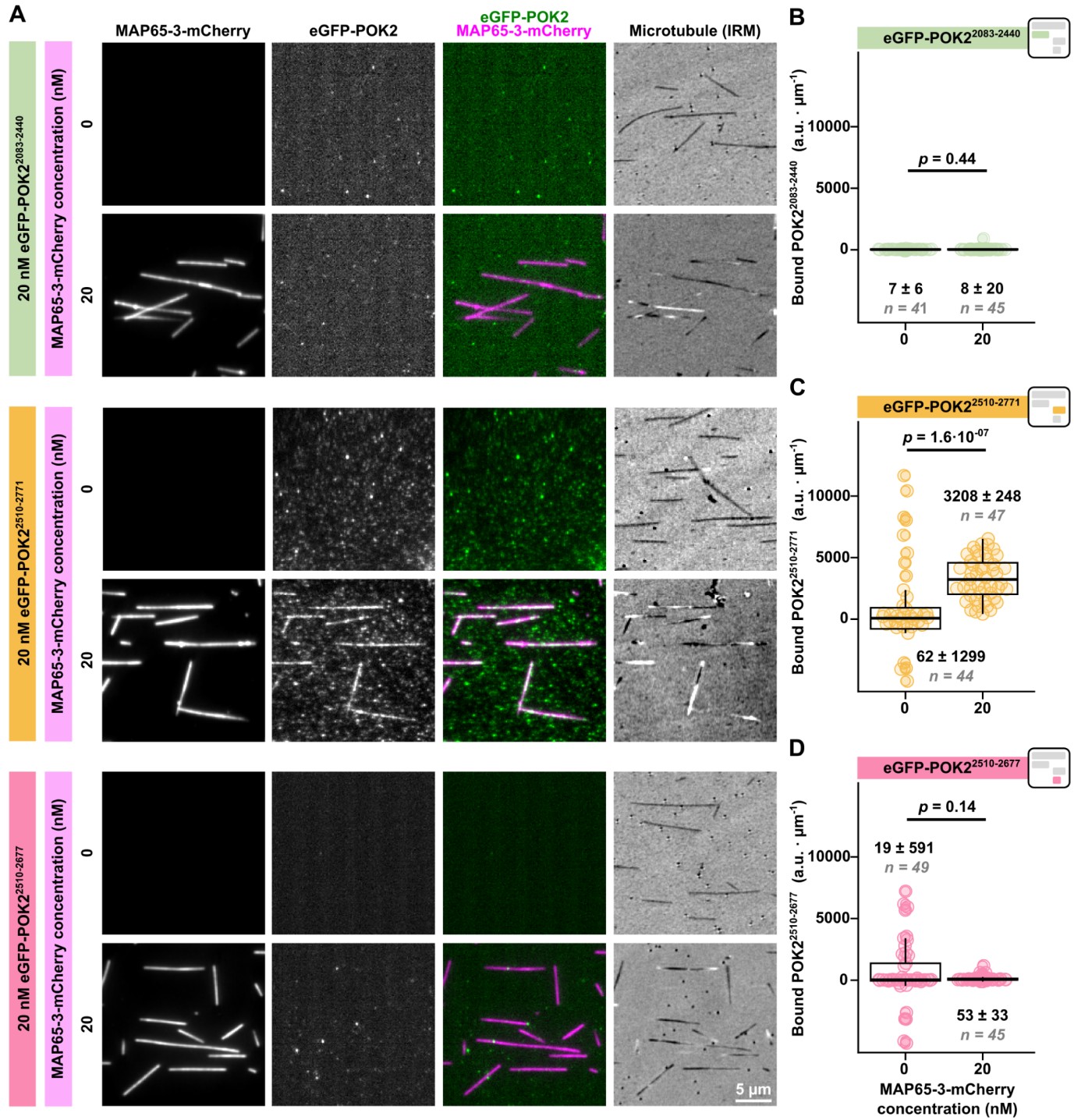

**Fig. 5. MAP65-3-dependent promotion of microtubule binding by the C-terminal domain of POK2 requires the final POK2 residues 2678–2771.** (A) TIRF microscopy images showing the binding of the indicated C-terminal eGFP–POK2 fragments (second column and green in third column) at 20 nM to microtubules (fourth column, black/white) in the presence (20 nM) or absence (0 nM) of equimolar MAP65-3–mCherry (first column and magenta in third column). Note that the eGFP–POK2$^{2510-2771}$ construct has a higher tendency to non-specifically stick to the glass surface compared to the other constructs (middle rows). (B–D) Box plots showing the binding by POK2 C-terminal fragments to microtubules (as shown in A). Median intensity per length values ±s.e.m. and number of particles are given. *P*-values were calculated from a Mann–Whitney–Wilcoxon *U*-test. Horizontal lines mark the median, boxes indicate the interquartile range and whiskers indicate the Tukey range. Square icons depict the colour schemes and arrangement of POK2 recombinant proteins as in Fig. 1A, and are placed here for easy reference. a.u., arbitrary units.

(Fig. 6A). The displacement analysis of co-diffusing traces identified two different populations with diffusion coefficients of 0.011±0.001 µm$^2$ s$^{-1}$ and 0.0013±0.0001 µm$^2$ s$^{-1}$ (mean±fit error, *n*=19 traces), being a factor of ~3 and ~30 smaller, respectively,

than that of the POK2 tail fragment alone (Fig. 6E,F; Fig. S4B,C). In summary, these data suggest that the C-terminal POK2 fragment and MAP65-3 directly interact to form complexes that can diffuse on microtubules.

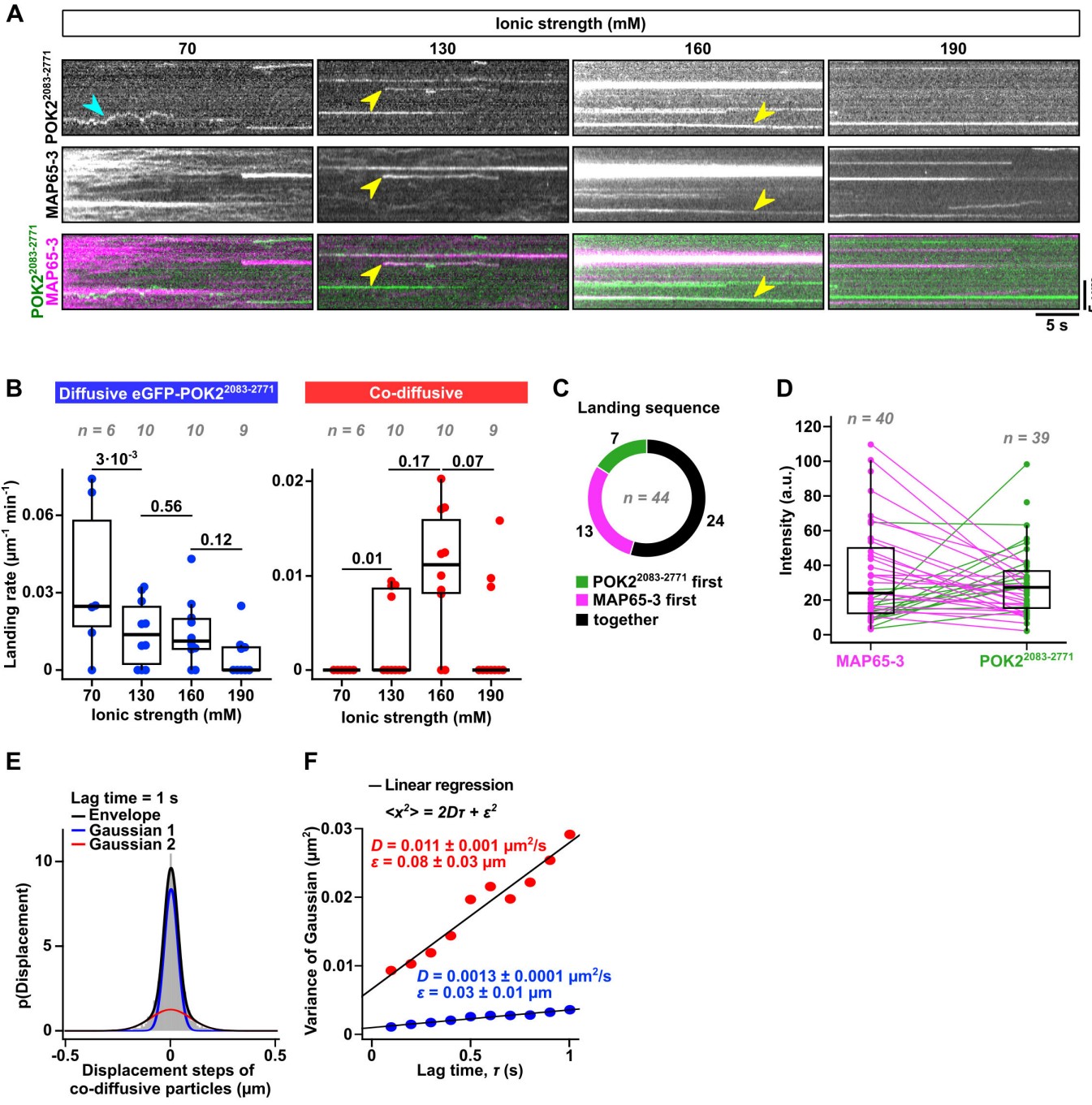

**Fig. 6. eGFP–POK2$^{2083–2771}$ and MAP65-3–mCherry co-diffuse on stabilised microtubules.** (A) Kymographs showing the binding and movement of 1 nM eGFP–POK2$^{2083–2771}$ and 0.5 nM MAP65-3–mCherry on stabilised microtubules at 70, 130, 160 and 190 mM ionic strength. Cyan arrowhead indicates diffusive eGFP–POK2$^{2083–2771}$. Yellow arrowheads indicate co-diffusive traces. (B) Landing rate of eGFP–POK2$^{2083–2771}$ diffusive events (left, blue) and co-diffusive events (right, red). As ionic strength increases, the landing rate of eGFP–POK2$^{2083–2771}$ decreases, whereas the co-diffusive traces peak at 160 mM. Numbers indicate $P$-values calculated using a Mann–Whitney–Wilcoxon $U$-test. Horizontal lines mark the median, boxes are the interquartile range, with whiskers marking the Tukey range. Grey numbers show the number of movies analysed, each with at least ten microtubules analysed for landing events. (C) Landing sequence of proteins that form diffusing POK2$^{2083–2771}$–MAP65-3 particles on the microtubule. (D) Paired intensity distribution of fluorescence intensities in diffusive POK2$^{2083–2771}$–MAP65-3 particles given as arbitrary units (a.u.) per protein. Boxes indicate the interquartile range, with whiskers marking the Tukey range. Horizontal lines within the box indicate median intensities. Magenta and green lines indicate intensity pairs of the same particle. Please note that the absolute values of eGFP–POK2$^{2083–2771}$ and MAP65-3–mCherry intensities cannot be compared directly. (E,F) Co-diffusive particles were tracked and analysed for their diffusion coefficient, $D$, and precision, $\varepsilon$, in the same manner as in Fig. 4F,G (see Fig. S4B,C and Materials and Methods for details). Values of $D$ and $\varepsilon$ are shown ±fit error in F. $n$=19 from three experiments.

## The C-terminal domain of POK2 directly binds lipids

Although our data show that the POK2 tail directly binds microtubules and that this interaction can be enhanced by MAP65-3, they do not explain how POK2 is retained at the CDZ after the PPB has disassembled and MAP65-3 has disappeared. Based on the high occurrence of positively charged amino acids in

the C terminus of POK2 (Fig. S1E), we hypothesised that POK2 accumulates at the CDZ due to direct interactions of the tail with negatively charged lipids in the plasma membrane. Similar interactions have been described for kinesin-1 (Anton et al., 2021; Wang et al., 2017) and kinesin-3 (Klopfenstein and Vale, 2004; Blatner et al., 2007; Hoepfner et al., 2005; Xue et al., 2010). We first screened for possible lipid–protein interactions using eGFP–POK2$^{2083–2771}$ in a lipid blot assay. We found that eGFP–POK2$^{2083–2771}$ bound to a variety of anionic lipids, such as phosphatidic acid (PA), cardiolipin (CL), and various phosphatidylinositol phospholipids, such as phosphatidylinositol 4-phosphate (PI4P), phosphatidylinositol (4,5)-bisphosphate [PI(4,5)P$_2$] and phosphatidylinositol (3,4,5)-trisphosphate [PI(3,4,5)P$_3$] (Fig. 7A; Fig. S6), suggesting that eGFP–POK2$^{2083–2771}$ might directly bind to the plasma membrane. To test the lipid-binding capacities of the C-terminal POK2 fragment in a more physiological setting, we prepared giant unilamellar vesicles (GUVs) of different lipid compositions, mixed them with 20 nM eGFP–POK2$^{2083–2771}$ and visualised them by highly inclined and laminated optical sheet (HILO) microscopy (Tokunaga et al., 2008). Consistent with our lipid blot assay data, eGFP–POK2$^{2083–2771}$ bound to and diffused on GUVs made of 75–100% PA and 50–100% CL but did not bind GUVs made of 100% dipalmitoylphosphatidylcholine (DOPC) (Fig. 7B; Movie 1; please note that some static patterns in the green channel for high PA concentrations are due to non-specific, surface-bound clusters; see Materials and Methods).

As the C terminus does not contain any known motifs or binding sites for lipids, we tested our truncation variants for lipid binding in the lipid blot assay. We found that eGFP–POK2$^{2510–2771}$ retained all lipid-binding capacities, whereas further truncations from the C-terminal end in eGFP–POK2$^{2083–2440}$ and eGFP–POK2$^{2510–2677}$ largely abolished lipid binding, with eGFP–POK2$^{2083–2440}$ having some residual binding activity to PA (Fig. 7A; Fig. S6). These findings suggest that the final region of the POK2 C terminus not only binds microtubules and MAP65-3 but also lipids in the plasma membrane at the CDZ.

## DISCUSSION
### A sequential and cooperative model for targeting of POK2 to the CDZ

Using an *in vitro* reconstitution approach, we identified the final ∼100 amino acids of the plant kinesin-12 POK2 as a versatile interaction hub that allows the kinesin to interact with (1) microtubules independently of the motor domain, (2) MAP65-3 and (3) anionic lipids *in vitro* (Fig. 8A,B). Similar truncation experiments *in vivo* and *in vitro* (Livanos et al., 2025) have also recently confirmed the existence of a second microtubule-binding site, mapped to amino acids 2503–2670. This region has been found to localise to microtubules *in vivo* and show a weak interaction with microtubules in an *in vitro* co-sedimentation assay. These findings align with our observations of the POK2$^{2510–2677}$ construct, which also retained some residual microtubule binding (Fig. 2A,B). Livanos et al. did not test the last ∼100 amino acids of POK2, which mediated the main microtubule-binding activity in our hands, but did show that this ∼100-amino-acid region binds to anionic lipids via electrostatic interactions based on lipid blot assays, like ours, and additional mutational analyses (Livanos et al., 2025). In addition, they identified a second, weaker lipid-binding interface (amino acids 2308–2330; Livanos et al., 2025). Consistent with their results, our POK2$^{2083–2440}$ construct, which contains this region, displayed residual lipid-binding activity in our lipid blot

assay – though with a different lipid specificity, likely due to the different amino acid range used. Based on our experiments and on the current literature on the *in vivo* function of POK2, we propose a sequential and cooperative model for the recruitment and stable deposition of POK2 at the CDZ (Fig. 8C) that warrants further testing *in planta*.

The ability of the POK2 tail to interact with multiple components might enable subcellular targeting to be more efficient, robust and specific. We propose that in preprophase, POK2 accumulates at cortex-proximal plus-ends of PPB microtubules through its own plus-end-directed motor activity (Fig. 8C step 1). *In vivo* experiments have suggested that this step is not essential for CDZ localisation but significantly increases its efficiency (Herrmann et al., 2018). The presence of a second microtubule-binding site in the tail of POK2 likely enhances the microtubule residency time and run length of the motor, similar to other kinesins with a second microtubule-binding domain in the tail (Stock et al., 2003; Weinger et al., 2011), thereby increasing the probability that the motor would reach the PPB and CDZ.

At the microtubule plus-ends, POK2 might be retained by direct interactions of its C-terminal domain with the microtubule lattice and/or MAP65 molecules, thereby accumulating the motor locally (Fig. 8C step 2, Fig. 8D). Consistent with this notion, POK2 constructs lacking the second microtubule-binding domain fail to localise to the PPB and CDZ, while still localising normally to the spindle and phragmoplast midzone (Livanos et al., 2025). Retention by MAP65 molecules might include the paralogues MAP65-1, MAP65-3 and MAP65-5, as all of them localise to the PPB (Smertenko et al., 2008) and interact with the POK2 tail *in vivo* (Herrmann et al., 2018). Although *in planta* experiments will have to clarify to what extent the single MAP65 paralogues might contribute to CDZ targeting of POK2, we will focus for the rest of the discussion on MAP65-3. It is the only MAP65 that yields a significant phragmoplast phenotype when knocked out (Caillaud et al., 2008; Müller et al., 2004; Ho et al., 2012) and has the strongest interaction with POK2 (i.e. via two distinct binding sites in the N- and C-terminal domains of POK2; Herrmann et al., 2018). Given that the stoichiometry of POK2–MAP65-3 complexes is variable and that both proteins interact directly with microtubules, the microtubule- and MAP65-3-binding by POK2 could be mutually exclusive or occur simultaneously. The interaction of the POK2–MAP65-3 heterocomplexes with microtubules was less sensitive to changes in the ionic strength than the interaction of POK2 alone with microtubules, suggesting that these interactions might be mediated by different binding interfaces and, thus, may occur simultaneously. For example, MAP65-3 might bind further upstream in the region of amino acids 2678–2726 (Fig. S1E), outside of the final ∼100-amino-acid region that mediates binding to the negatively charged microtubules. More refined *in vitro* and *in planta* experiments in the future are required to dissect this triangle of interactions.

Plus-end retention and enrichment through MAP65-3, subsequently, would then grant sufficient time for the offloading of POK2 onto lipids with anionic headgroups within the cell membrane, anchoring POK2 at the future CDZ for efficient phragmoplast positioning (Fig. 8C step 3). There is no evidence that MAP65-3 will contribute directly to the cortical anchoring of POK2, given that MAP65-3 leaves the cortex after prophase along with the PPB microtubules (Müller et al., 2004; Ho et al., 2011) and POK2 localisation is unchanged in MAP65-3 mutants (Herrmann et al., 2018). We expect that plus-end targeting (Fig. 8C step 1) and offloading (Fig. 8C step 3) still occur during

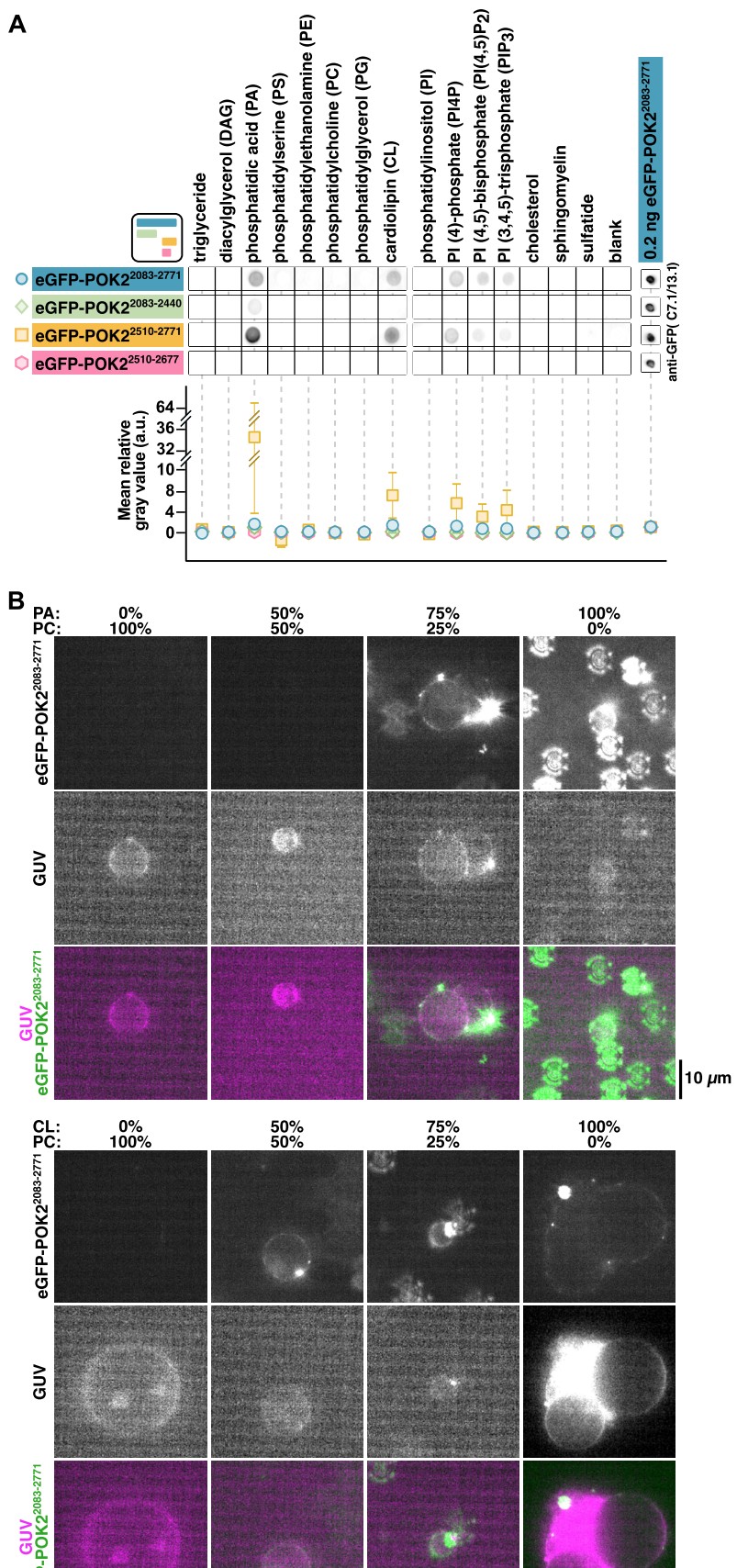

**Fig. 7. C-terminal fragments of POK2 bind anionic lipids directly.** (A) Lipid blot assay with the indicated eGFP–POK2 fragments and immobilised lipids probed with anti-GFP antibodies. Plot below shows the quantification of the respective signals normalised to the individual loading controls (i.e. 0.2 ng eGFP–POK2$^{2083–2771}$) directly dotted onto the nitrocellulose membrane. Data points are mean ±s.e.m. from three repeats (a.u., arbitrary units). Square icon depicts the colour scheme and layout of POK2 recombinant proteins in Fig. 1A. (B) Fluorescence images of eGFP–POK2$^{2083–2771}$ binding to GUVs made from the indicated ratios of PA and phosphatidylcholine (PC; top panels), or of CL and PC (bottom panels). GUVs were unlabelled and visualised with HILO microscopy when illuminated by reflections from fluorescent contamination. Images shown are representative of three experiments.

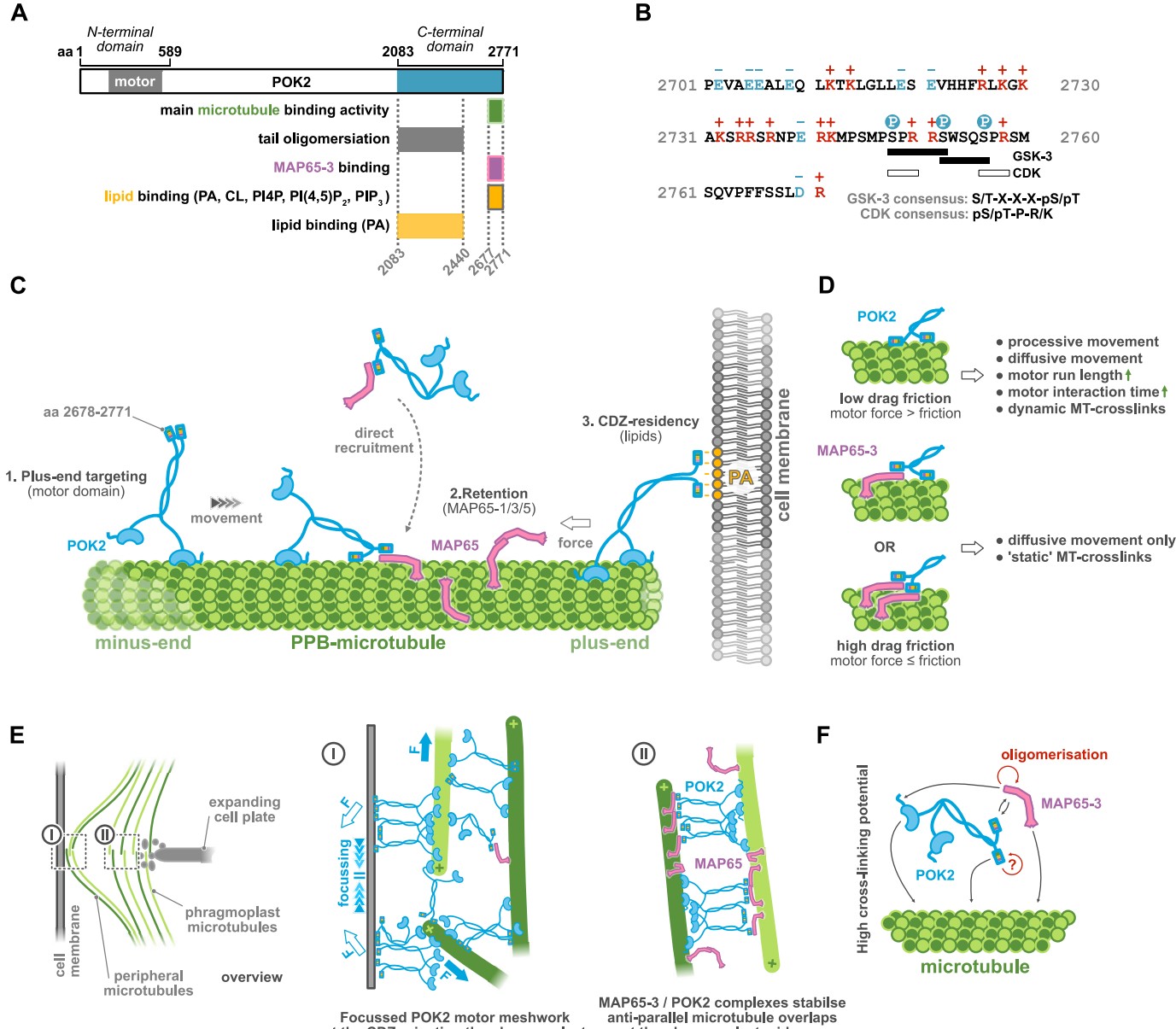

**Fig. 8. Role of the C-terminal interaction hub in POK2 functionality at the CDZ and at the phragmoplast midzone.** (A) Graphical summary of the binding interfaces identified in this study. aa, amino acid; PIP$_3$, PI(3,4,5)P$_3$. (B) The final sequence of the POK2 C-terminal tail is rich in amino acids with positively charged side chains (red) and poor in negatively charged amino acids (blue). It further features kinase consensus sites of kinases that are known to phospho-regulate microtubule association of kinesins. White P on blue circle, phosphorylation. (C) Sequential model of POK2 targeting and immobilisation at the PPB and CDZ involving the C-terminal interaction hub and its interactions with MAP65 proteins (MAP65-1/3/5) and lipids in the cortical cell membrane (see Discussion for a detailed explanation). (D) Depending on the nature of interaction of the POK2 interaction hub with the microtubule, we expect the motor to experience low drag friction that would increase its interaction time and run length, allowing processive and diffusive movement as well as dynamic microtubule crosslinks (no MAP65; upper panel) or high drag friction (in the presence of MAP65; middle and lower panels) only allowing diffusive movement and rather static crosslinks. MT, microtubule. (E) Scheme showing the different tasks of POK2 at the phragmoplast midzone (I) and the CDZ (II) during phragmoplast orientation. At the phragmoplast midzone (I, middle panel), heterocomplexes of POK2 and MAP65-3 create stable static crosslinks in antiparallel microtubule overlaps of the interdigitating phragmoplast halves. Please note that the POK2–MAP65-3 heterocomplex can span larger distances than MAP65-3 dimers (i.e. ~30 nm for its human homologue PRC1; Subramanian et al., 2013) due to a POK2 contour length of ~300 nm. At the CDZ (II, right panel), membrane-bound POK2 walks to the plus-ends of peripheral microtubules, thereby creating pushing forces on both phragmoplast poles, orienting the phragmoplast. Counter forces focus the diffusively tethered C-terminal POK2 domains on the cell membrane. In the overlap of interdigitating peripheral microtubules, a meshwork of motile POK2 motors crosslinks the overlapping microtubules and maintain a defined overlap length, preventing microtubule sliding past the focal point and thus CDZ de-focussing. F, force. (F) Scheme indicating the interaction between POK2, MAP65 and microtubules, granting heteromeric complexes a high crosslinking potential.

phragmoplast positioning on peripheral microtubules (Chugh et al., 2018). In this manner, the cortical pool of POK2 could also be replenished in the absence of MAP65-3, although at reduced efficiency. Although this model needs to be thoroughly tested *in vivo*, the *in planta* data recently published by Livanos et al. (2025)

already support central aspects of this model confirming that targeting to the PPB and CDZ depends on the second microtubule-binding domain and that long-term retention at the CDZ requires the interaction of the POK2 tail with anionic lipids in the cortical membrane.

## Long-term retention of POK2 at the CDZ

Our data show that the POK2 tail specifically binds to anionic lipids like PA, PI4P, PI(4,5)P$_2$, PI(3,4,5)P$_3$ and CL. Although CL has only been found on mitochondria and plastid membranes (Zhou et al., 2016), enrichment of some of the other lipids at the CDZ might act as a cue for the selective binding and long-term retention of POK2 at the CDZ. The cell membranes at animal and fungal cell division sites, for example, are enriched in PA, phosphatidylethanolamine and phosphatidylserine (Caillaud, 2019; Field et al., 2005; Kouranti et al., 2006), and are depleted of PI(4,5)P$_2$ (Di Paolo and De Camilli, 2006). This depletion is required to destabilise F-actin for cytokinetic abscission, as several F-actin nucleators and stabilisers bind PI(4,5)P$_2$ (Di Paolo and De Camilli, 2006; Golub and Pico, 2005; Pollard and Borisy, 2003; Yin and Janmey, 2003). In plants, there is no known lipid that specifically marks the CDZ. Although there is an actin depletion zone at the CDZ, PI(4,5)P$_2$ uniformly decorates the plasma membrane within the cell division plane (Simon et al., 2016). Consistent with non-specific lipid retention, a decrease in the lipid specificity of POK2, by swapping its binding domain with that of the mammalian small GTPase KRAS4B, which binds a broader spectrum of lipids, does not alter POK2 CDZ localisation *in vivo* (Livanos et al., 2025). Hence, there must be an additional mechanism that prevents POK2 from diffusing away from the CDZ onto the remaining cell membrane. For example, there could be an additional interaction partner at the CDZ.

One candidate interaction partner for long-term retention of POK2 at the CDZ would be the microtubule- and CDZ-associated TANGLED (TAN). TAN is a PPB- and CDZ-resident protein that physically interacts with the POK2 paralogue POK1, thereby retaining it at the PPB and CDZ (Walker et al., 2007; Lipka et al., 2014; Mills et al., 2022). Efficient POK1 retention further requires AIR9, a MAP that is functionally redundant to TAN, at the PPB and CDZ (Mills et al., 2022). However, AIR9 alone cannot support POK1 retention throughout the cell cycle, as it dissociates from the CDZ upon PPB disassembly and reappears only when the phragmoplast contacts the cell membrane (Buschmann et al., 2006, 2015). Given their redundancy, TAN is proposed to also recruit and retain POK2 to the PPB and CDZ, but physical interaction between TAN and POK2 has not been shown. Furthermore, CDZ localisation of TAN and of POK1 and POK2 is mutually interdependent, as TAN is lost from the CDZ in *pok1 pok2* double mutants (Walker et al., 2007; Lipka et al., 2014), making it unlikely that TAN is an initial CDZ anchoring factor for POK2.

The IQ67 DOMAIN family member 8 (IQD8) might be another candidate to further restrict POK2 localisation to the CDZ. IQD8 localises throughout mitosis to the cell membrane next to the PPB and CDZ independently of microtubules and directly interacts with the C-terminal fragment of the POK2 tail (Kumari et al., 2021). However, the distribution of IQD8 on the cell membrane is broader than that of other CDZ components (Kumari et al., 2021). Therefore, there must be an additional mechanism that keeps POK2 focussed to the CDZ. Focussing might be achieved through peripheral phragmoplast microtubules once they encounter the membrane-bound POK2. A continuous interaction would keep POK2 from diffusing away, while forces generated by POK2 on those microtubules create counter pushing forces that focus the motor population to a narrow band at the CDZ (Fig. 8E) (Chugh et al., 2018). Although mechanical forces could explain focussing, it is unclear how over- or de-focussing could be prevented.

## Maintaining a focussed CDZ

Once the division zone has narrowed, maintaining it becomes challenging during the radial expansion of the phragmoplast. The expansion leads to continuous changes in the geometry of the peripheral microtubules towards more shallow angles between microtubules and the cell membrane (Sasaki et al., 2019). Shallow angles likely favour the interdigitated sliding of peripheral microtubules from opposite phragmoplast halves along the plasma membrane that – together with microtubule polymerisation – would defocus the POK2 membrane-bound pool again if there were no further interactions. Since POK2 contains a second microtubule-binding domain and motors probably form multimers, we propose that a second, non-membrane-bound pool of POK2 motors crosslinks peripheral microtubules that emanate from opposing sides of the phragmoplast (Fig. 8E, F panel I). In these antiparallel overlaps, a meshwork of entangled POK2 motors could – in analogy to the minus-end-directed human kinesin-14 – maintain a stable overlap size, preventing POK2 defocussing. Kinesin-14 is a weak (∼0.5 pN, as compared to ∼0.3 pN for POK2; Liu et al., 2024; Chugh et al., 2018) minus-end-directed motor with a second ATP-independent microtubule-binding domain in its tail that crosslinks antiparallel microtubules in the animal spindle and drives their inward sliding by (1) active power strokes sliding one microtubule along the other (Liu et al., 2024; Braun et al., 2017; Reinemann et al., 2018), (2) entropic effects of diffusing motors trying to maximise microtubule overlap in a concentration-dependent manner (Braun et al., 2017), and by (3) restricting outward-directed forces generated by kinesin-5 (Reinemann et al., 2018). Transferring these mechanical concepts to the antiparallel overlap of peripheral microtubules, we would expect POK2 to drive extensile sliding of antiparallel microtubules, thus reducing the overlap, while entropic effects by POK2 would maintain a minimal, focussed overlap. A similar function of stably maintaining an antiparallel microtubule overlap of a defined length might be carried out by the POK2 subpopulation at the phragmoplast midzone. Here, POK2 together with MAP65-3 (Müller et al., 2004; Ho et al., 2011) maintains the engagement of the phragmoplast halves that interdigitate with their plus-ends at the midzone (Fig. 8E panel II). In contrast to the more dynamic situation at the peripheral overlaps, we expect crosslinks formed by POK2 and MAP65-3 to be rather static (Fig. 8D–F). In this context, we estimated the relative binding strength of the motor domain and that of the tail microtubule domain in the absence or presence of MAP65-3. Based on our measured diffusion coefficient $D \approx 0.011~\mu m^2~s^{-1}$ of the complex and the POK2 motor speed $v \approx 0.43~\mu m~s^{-1}$ (Chugh et al., 2018), we estimate the frictional drag force required to slide apart the overlap to be $F = \dfrac{v k_B T}{D} \approx 0.11~pN$ (Table S4), where $k_B$ is the Boltzmann constant and $T$ the absolute temperature (Bormuth et al., 2009). However, as the diffusion coefficient can be ten times smaller (Fig. 6F), the frictional force can reach 1.2 pN, fourfold larger than the maximum motor force of ∼0.3 pN (Chugh et al., 2018) (Fig. 8D,E; Table S4). Thus, the motility of such complexes would be significantly hampered. This impedance might be further increased by the second interaction of MAP65-3 with the N-terminal extension of the POK2 motor (Herrmann et al., 2018), although it is currently not clear whether both MAP65-3-binding sites on POK2 can be occupied at the same time. Without MAP65-3, if both microtubule-binding sites of POK2 are interacting with the microtubule lattice, we would not expect a significant speed reduction of the motor as the tail friction would only be ∼10% of the maximum motor force (Table S4). Thus, we expect that full-length motors by themselves can translocate to

microtubule plus-ends, whereby the second microtubule-binding site in the POK2 tail should enhance the processivity and run length.

## Regulation of POK2 activity

In summary, our data and that of others indicate that nearly all interactions of POK2 with components of the PPB and CDZ (such as microtubules, MAP65-3 and lipids in the cell membrane), or of the phragmoplast midzone (such as microtubules and MAP65-3), are mediated by the same last ~100 amino acids in the C-terminal domain of POK2. Future work will have to establish whether these interactions might occur simultaneously or whether they are mutually exclusive.

Mutually exclusive interactions of the POK2 tail might be with microtubules and lipids. The interaction of non-motor microtubule-binding domains of MAPs commonly relies on electrostatic interactions between positive charges within the microtubule-binding domain and negative charges within the E-hooks in the tubulin C termini exposed on the microtubule lattice (Drechsler et al., 2019). In the unstructured C-terminal end of the POK2 tail, there is an accumulation of amino acids with positively charged side chains in the region of amino acids 2710–2760 that appear crucial for microtubule binding, as only the POK2$^{2510–2771}$ construct, and not the POK2$^{2510–2677}$ construct, binds microtubules *in vitro*. Our observation that the full C-terminal construct (amino acids 2083–2771) appears to be necessary to reach microtubule affinities in the nanomolar range might be explained by the contribution of additional positive charges and/or the presence of coiled-coil stretches that create bivalency through dimerisation or tetramerisation, thereby increasing microtubule affinity (Drechsler et al., 2019). The tail of the functionally redundant and substantially shorter orthologue of POK2, POK1, seems to lack such a second microtubule-binding domain (Fig. S7), suggesting that it cannot crosslink microtubules on its own. It therefore must rely on its interaction with either TAN or AIR9 (Mills et al., 2022) for microtubule crosslinking, given that an interaction with any of the MAP65 paralogues has not been described yet. Although this hypothesis must be tested by future *in vitro* reconstitution experiments with purified components, it is consistent with *in vivo* data suggesting that PPB microtubule localisation of POK1 relies on its interaction with the microtubule- and CDZ-associated TAN (Walker et al., 2007; Lipka et al., 2014), and provides a possible explanation for its absence from the antiparallel overlaps at the phragmoplast midzone (Lipka et al., 2014). Altogether, our results suggest that the larger POK2 incorporates all important interactions required for CDZ localisation and phragmoplast orientation (i.e. motor activity, a second microtubule-binding domain for crosslinking, as well as membrane binding), whereas the shorter POK1 relies on additional factors like TAN, AIR9 or other still-unknown factors to mediate microtubule crosslinking and CDZ residency.

Given that POK2 interacts only with anionic lipids, we suggest that the same region of charged side chains is essential for binding to microtubules and lipids. Under these circumstances, the plant cell needs another level of control to selectively allow microtubule (phragmoplast midzone) or cell membrane (CDZ) association – potentially by post-translational modifications. Indeed, the charged region in the POK2 tail also contains consensus sites recognised by cyclin-dependent kinases (CDKs; i.e. pS/pT-P-R/K) and glycogen synthase kinase 3-like kinases (GSK-3s; i.e. S/T-X-X-X-pS/pT) (Fig. 8B). These kinases commonly control the microtubule association of MAPs and motors in a cell cycle-dependent manner and are suggested to also phosphorylate other components

of the CDZ such as IQ67 DOMAIN proteins (Dahiya and Bürstenbinder, 2023). For POK2, we would expect CDK- and/or GSK-3-dependent phosphorylation to reduce the microtubule affinity of the second microtubule-binding domain in cell cycle phases outside cytokinesis to reduce detrimental distortions of their microtubule cytoskeleton through unwanted microtubule bundling by POK2. Other proline-guided kinases like the mitogen-activated protein kinases (MAPKs) that, for example, regulate the microtubule association of MAP65 paralogues during cytokinesis (Smertenko et al., 2006), in contrast, are unlikely to target these sites in POK2. The S-P sites in question do not satisfy the general P-X-pS/pT-P consensus for MAPKs (Gonzalez et al., 1991), and although the first CDK consensus site (i.e. M-P-S-P) would also qualify for the less commonly used Φ-X-pS/pT-P MAPK consensus site (Miller and Turk, 2018), a proline directly preceding the S-P site greatly reduces MAPK phosphorylation at this site (Gonzalez et al., 1991).

Overall, the interactive hub of the POK2 tail together with its motor activity and biochemical regulation may explain its complex spatiotemporal localisation, may contribute to the mechanical patterning of cells by focussing and defining the cell plate fusion site, and may *in planta* be not only an interaction hub but also a connectivity hub.

## MATERIALS AND METHODS
### Purification of POK2 C-terminal constructs

cDNA of *Arabidopsis thaliana* POK2 (At3g19050; gifted from Sabine Müller, Friedrich-Alexander-Universität Erlangen-Nürnberg, Erlangen, Germany) encoding residues 2083–2771, 2083–2440, 2510–2771 and 2510–2677 was cloned into pFastBac1 (Bac-to-Bac system, Gibco) and fused to an N-terminal eGFP with GSAGSAAGSG linker and C-terminal 8×His-tag with GSGSGSG linker (Tables S1,S2). Plasmids were transformed into DH10Bac *Escherichia coli* (Gibco, 10361012) to be taken up by its bacmid. Successful transformants were selected with blue–white screening and cultured, from which their bacmids were purified. ExpiSf insect cells (Gibco) were then transfected with purified recombinant bacmids to obtain virus. A fresh culture of ExpiSf cells was then transfected with the virus to induce protein expression. A 30 ml volume of transfected cells expressing protein were pelleted and resuspended in 10 ml cold Lysis Buffer [50 mM sodium phosphate buffer pH 7.5, 300 mM KCl, 1 mM MgCl$_2$, 10% glycerol, 0.1 mM ATP, 5 mM β-mercaptoethanol, 1 mM phenylmethylsufonyl fluoride (PMSF), and 1× cOmplete™ EDTA-free Protease Inhibitor Cocktail (Roche) and 25 U/ml Universal Nuclease for Cell Lysis (Pierce, Thermo Fisher Scientific)] and sonicated at 75% power with 10 s pulses for a total of 5 min with 40 s pause time in between pulses (Active Motif Q120AM). Lysed cells were clarified by 7745 *g* centrifugation at 4°C for 20 min. The supernatant was then passed through a 0.22 μm MCE filter membrane (Merck) and loaded onto a 5 ml HiTrap TALON Crude column (Cytiva) using the Äkta system (Cytiva). The column was then washed with 20 ml Wash Buffer (Lysis Buffer but without nuclease) and 50 ml Desalting Buffer (Lysis Buffer but without PMSF, protein inhibitors or nuclease). The sample was eluted as fractions with 10 ml of 100 mM, 15 ml of 300 mM, and finally 30 ml of 500 mM imidazole in the Desalting Buffer. Peak fractions (in total 1.5 ml) were pooled and loaded onto a 5 ml HiTrap Desalting column (Cytiva) and eluted using 15 ml Desalting Buffer. Proteins were verified by SDS-PAGE with Coomassie staining and anti-GFP (Roche, 11814460001; RRID: AB_390913; 1:2000) and anti-His (Invitrogen, MA1-21315; RRID: AB_557403; 1:800) for western blotting (Figs S1A,B and S8C,D). Oligomerisation of proteins was also verified by blue native PAGE using Novex WedgeWell 4–12% Tris-Glycine Gels (Invitrogen, XP04122BOX) with 100 ng of each protein per lane and were assessed with anti-GFP (Roche, 11814460001; RRID:AB_390913; 1:2000) western blotting and silver staining (Pierce, 24612) (Figs S1C and S8E–G). Please note that much of the eGFP–POK2$^{2083–2771}$–8×His protein remained in the pellet after lysis

and clarification. Nevertheless, there was enough soluble protein for purification, yielding ~100 µg protein per 30 ml culture. Plasmids for POK2 tail constructs are available on request.

## Purification of MAP65-3

The expression plasmid for *Arabidopsis thaliana* MAP65-3 (At5g51600), MAP65-3–mCherry–8×His in pFastBac1, was generated by GenScript and transfected into ExpiSf insect cells for expression of recombinant MAP65-3 as described for POK2 tail constructs (Table S1). Cell pellets (15 ml) expressing MAP65-3–mCherry–8×His were resuspended in 10 ml cold Lysis Buffer [50 mM sodium phosphate buffer pH 7.9, 100 mM NaCl, 0.5% Triton X-100, 30 mM imidazole, 0.1 mM ATP, 5 mM β-mercaptoethanol, 1 mM PMSF, 1× cOmplete™ EDTA-free Protease Inhibitor Cocktail (Roche) and 25 U/ml Universal Nuclease for Cell Lysis (Pierce, Thermo Fisher Scientific)] and sonicated in the same manner as for the POK2 constructs. The supernatant was then passed through a 0.8 µm cellulose acetate membrane (Whatman, FP30/0.8 10462240) and loaded onto a 1 ml HisTrap HP column (Cytiva) using the Äkta system (Cytiva). The column was washed with 20 ml Wash Buffer (Lysis Buffer but without PMSF and Triton X-100) and then eluted using 5 ml of 150 mM, 5 ml of 300 mM and 10 ml of 500 mM imidazole in Desalting Buffer (Lysis Buffer but without PMSF and Triton X-100 and with 5% glycerol). Peak fractions (in total 1.5 ml) were then loaded on a 5 ml HiTrap Desalting column (Cytiva) and eluted using 15 ml Desalting Buffer. The samples were then verified by SDS-PAGE with Coomassie staining and with anti-RFP (Chromotek, 6G6-100; RRID:AB_2631395; 1:2000) and anti-His (Invitrogen, MA1-21315; RRID:AB_557403; 1:800) western blotting (Figs S4A and S8I). Plasmid for MAP65-3–mCherry–8×His is available on request.

## GMPCPP- and paclitaxel-stabilised microtubules

Porcine brain tubulin (0.3 mg/ml; own preparation according to Castoldi and Popov, 2003) was mixed with 1 mM GMPCPP (Jena Bioscience, NU-405S), 1 mM $MgCl_2$ in PEM (80 mM K-PIPES pH 6.9, 1 mM EGTA, 1 mM $MgCl_2$) in a total volume of 100 µl, incubated on ice for 5 min, then at 37°C for 1 h, pelleted down using 22 p.s.i. (~150 kPa) pressurised air for 5 min at room temperature [Beckman Coulter; 340401 (Airfuge), 347595 (A-95 rotor)] or at 20,000 *g* 20 min at 37°C. The supernatant was discarded, and the pellet was resuspended in 200 µl PEM supplemented with 10 µM paclitaxel (Sigma Aldrich, T7191) and kept at 37°C until ready for use. For labelled microtubules, rhodamine-labelled tubulin (Hirst et al., 2020) was pre-mixed with unlabelled tubulin to 10% before polymerisation.

## Subtilisin-treated GMPCPP- and paclitaxel-stabilised biotinylated microtubules

GMPCPP- and paclitaxel-stabilised microtubules were grown as described above from 0.3 mg/ml porcine brain tubulin, of which 20% were biotinylated. After centrifugation, the microtubule pellet was resuspended in PEM, supplemented with 10 µM paclitaxel and 0.2 mg/ml subtilisin A (Sigma Aldrich, P5380), and incubated at 30°C for 2 h. To stop the enzyme reaction, 10 mM PMSF was added. The reaction was then centrifuged at 22 p.s.i. for 5 min at room temperature [Beckman Coulter; 340401 (Airfuge), 347595 (A-95 rotor)], and the pellet was resuspended in PEM containing 10 µM paclitaxel and kept at 37°C until ready for use. The subtilisin treatment was verified by SDS-PAGE with Coomassie staining (Figs S1D and S8H).

## *In vitro* microtubule binding assay

Two pieces of ~100 µm thick double-sided tape were placed on a 22×22 mm$^2$ hexamethydisilane (HMDS; Sigma Aldrich, 379212) glass cover slip (Wedler et al., 2022) to create a thin 1 mm-wide channel. A second 18×18 mm$^2$ HMDS glass cover slip was placed on top, thus sandwiching the tape in between to create a channel where there was no tape. Reagents were flowed into the channel under vacuum or drawn across the channel with filter paper in this order: step 1, 60 µl deionised water; step 2, 20 µl PEM; step 3, 20 µl anti-β-tubulin monoclonal antibody (100-fold diluted in PEM; Sigma Aldrich, T7816), incubated at room temperature for 3 min; step 4, 20 µl PEM; step 5, 20 µl 1% Pluronic F-127 (Sigma Aldrich,

P2443) in PEM and incubated at room temperature for 10 min; step 6, 20 µl PEM; step 7, 20 µl stabilised microtubules 1:1 diluted in PEM; step 8, 20 µl PEM; step 9, 20 µl of desired protein(s) in Reaction Buffer [0.16 mg/ml casein, 20 mM D-glucose, 20 µg/ml glucose oxidase, 8 µg/ml catalase, 1 mM ATP, 5 mM dithiothreitol (DTT) in PEM20 (20 mM K-PIPES pH 6.9, 1 mM $MgCl_2$, 1 mM EGTA)]. For the final Reaction Buffer, PEM20 (20 mM K-PIPES) rather than PEM (80 mM K-PIPES) was used to compensate for the high salt concentration of the eGFP–POK2$^{2083-2771}$ storage buffer. Microtubule binding assays with eGFP–POK2 constructs on microtubules labelled with 10% rhodamine were imaged with a custom-built TIRF setup (Simmert et al., 2018; Chugh et al., 2018) where both rhodamine and eGFP were excited by the 488 nm laser. Due to the use of a broad filter in the green channel, there was bleed-through of the eGFP signal into the rhodamine channel (see Figs 1B and 2A).

Microtubule binding assays with eGFP–POK2$^{2083-2771}$ and MAP65-3–mCherry on unlabelled microtubules were performed similarly. Of note in step 9, the desired concentrations of eGFP–POK2$^{2083-2771}$ and MAP65-3–mCherry were mixed directly in the reaction buffer. For assays varying ionic strength (Fig. 6A,B), KCl was used in the final reaction buffer. Assays were imaged with custom-built dual-colour TIRF and IRM setup similar to that previously described (Chugh et al., 2018; Schellhaus et al., 2017; Wedler et al., 2022).

Microtubule binding assays with subtilisin-treated and untreated microtubules were performed similarly but used instead anti-Biotin antibodies (diluted 100-fold in PEM; Invitrogen, 03-3700; RRID: AB_2532265) to immobilise the microtubules to the surface, as subtilisin treatment removes the tails to which the anti-β-tubulin antibody binds.

Images were analysed using FIJI (Schindelin et al., 2012). Segmented lines were drawn along randomly selected microtubules, avoiding ends, crossovers and bundles in the microtubule-only channel. These lines were saved as regions of interest (ROIs). The lines were then moved to a nearby microtubule-absent region and saved as background ROIs. The raw integrated density from the protein channel and length were measured from these lines for both ROIs. Binding was calculated by subtracting the raw integrated density of background ROIs from the microtubule ROIs and divided by the length measured. Data were pooled from two to four experimental repeats, performed on different days, using different flow cells for each condition. For each condition, three to five images were analysed, wherein at least ten microtubules for each image were analysed.

## *In vitro* microtubule single-molecule assay

The experimental setup was the same as for the *in vitro* microtubule binding assay, just at concentrations of eGFP–POK2$^{2083-2771}$ where single molecules were visible [2 nM in a PEM20-based Reaction Buffer (0.16 mg/ml casein, 20 mM D-glucose, 20 µg/ml glucose oxidase, 8 µg/ml catalase, 1 mM ATP, 5 mM DTT)]. For experiments additionally with MAP65-3–mCherry, 1 nM eGFP–POK2$^{2083-2771}$ and 0.5 nM MAP65-3–mCherry were used. In this case, unlabelled, stabilised microtubules were used and visualised using IRM (Simmert et al., 2018). Time-lapse imaging with 100 ms exposure and frame time was performed using the custom-built TIRF setup (Wedler et al., 2022; Schellhaus et al., 2017; Chugh et al., 2018). Depending on the setup, the pixel size was 105.1 nm (Figs 1–3) or 148.5 nm (Figs 4–7). Data shown in Fig. 3 were pooled from analysing five time-lapse image series from two experiments in different flow channels performed on the same day. Data shown in Fig. 6 were pooled from analysing three or four time-lapse image series per condition, with each condition in a different flow cell. The sets of experiments varying ionic strength were performed three times on different days.

## Intensity analyses

Kymographs were drawn along microtubules to obtain traces of single-molecule microtubule-bound events. To obtain the particle intensities, particle traces were scanned for the first ~40 frames along segmented lines using FIJI (Schindelin et al., 2012). Retrieved intensity values were corrected for the local background next to the particle trace and averaged to compensate for frame-to-frame fluctuations in particle intensity (caused, for example, by movement of the microtubule out of the TIRF plane or intensity

fluctuation of the light source). To obtain bleaching steps, particle traces were manually fitted with segmented lines in FIJI yielding line scans that were locally background corrected. Bleaching steps within a single trace were identified manually, and flanking graph sections were fitted using Origin (Pro) 2024 (OriginLab Corp., USA) for a regression line with slope=0 to obtain an intensity average for the single sections.

### Landing rate analyses
Diffusive traces were identified on kymographs drawn along microtubules and counted using FIJI (Schindelin et al., 2012). Landing rate was calculated as the number of events divided by the length of microtubule analysed and the duration of the kymograph.

### Interaction time analyses
Diffusive (Fig. 3B) and stationary traces (Fig. S3B) on microtubules and background traces (Fig. S3A) from kymographs drawn next to microtubules were traced using FIJI (Schindelin et al., 2012) segmented lines. Only events where molecules associated and dissociated during the kymographs were considered. The survival plots of the interaction time distributions were fitted using Origin (Pro) (OriginLab Corp., USA).

### Bleaching probability analyses
The interaction time of diffusive traces from four time-lapse movies were measured in FIJI (Schindelin et al., 2012). This dataset includes all traces, regardless of whether molecules associated or dissociated during the movie duration. Bleaching probability was calculated by dividing the number of bleaching events by the total interaction time (in frames). Intensity drops to background levels have not been considered in this analysis, as we cannot formally discriminate between the bleaching of the last lit GFP or a particle dissociating from the microtubule lattice.

### Diffusion coefficient analyses
Images were analysed for traces using FIESTA (Ruhnow et al., 2011). Traces were checked manually to determine whether they were properly connected or disconnected and whether there were missing points in the FIESTA traces. From the tracked traces, the $x$- and $y$-positions were exported and rotated such that the $x$-position corresponded to the position along the microtubule axis. Distributions of displacements for lag times ($\tau$=0.1–2.0 s, sampled at 0.1 s intervals; representative distributions shown in Figs S3D and S4C) were fitted using Gaussian Mixture Models with K-means clustering for initialisation. The Akaike Information Criterion (AIC) and Bayesian Information Criterion (BIC) were used to assess the goodness of fit and determine the optimal number of components (Figs S3C and S4B). Based on the largest reduction in AIC and BIC, a two-Gaussian model provided the best fit to the data. The variance of the Gaussians was plotted against the lag time $\tau$, resulting in a mean squared displacement (MSD) plot. We fitted the line $MSD = 2D\tau + \varepsilon^2$ to the data to determine the diffusion coefficient, $D$, and tracking precision, $\varepsilon$ (Bormuth et al., 2009), where we assumed one-dimensional diffusion along the long axis of the microtubule.

### Lipid blot assay
Membrane lipid strips (Echelon Biosciences, P-6002) were first blocked with phosphate buffered saline (PBS) supplemented with 0.1% Tween20 (PBS-T) with 3% bovine serum albumin (BSA; Carl Roth, 0052.2) overnight at 4°C or at room temperature for 1 h with gentle shaking. The strips were then placed in 0.2 μg/ml protein diluted in PBS-T with 3% BSA and incubated at room temperature for 1 h with gentle shaking. Next, the strips were treated in the same manner as for anti-GFP (Roche, 11814460001; RRID:AB_390913; 1:2000) western blotting with washing steps using PBS-T. Data shown in Fig. 7A were pooled from three experimental repeats performed on different days, with individual lipid blots and plots shown in Fig. S6. See also Fig. S8B.

### GUV assay
Lipid mixtures of 10, 50, 75 or 100% (mol) phosphatidic acid (Avanti, 840101) or cardiolipin (Avanti, 840012) were prepared in the remaining molar percentage of dipalmitoylphosphatidylcholine (DOPC) (Avanti,

850375) to a total lipid concentration of 1 mg/ml in chloroform (Sigma, 288306). Then, 20 μl 1% polyvinyl alcohol (PVA) was spread on a microscope slide and desiccated under vacuum for 10 min. Next, 10 μl of the lipid mixture was spread over the PVA-coated surface and desiccated under vacuum for another 10 min. Silicon rings of 15 mm inner diameter were sealed on to the lipid–PVA surface with vacuum grease. Then 50 μl of 450 mM sucrose was gently added over the lipids and incubated at room temperature for 1 h to form giant unilamellar vesicles (GUVs). Hydrated GUVs were then collected into a LoBind tube (Eppendorf, 0030108116). To prepare the sample for imaging, a pre-cut 4×4-well cell culture gasket (Grace Bio-Labs, 103250) was placed onto a 22×22 mm² HMDS (Wedler et al., 2022) glass cover slip. Next, 5 μl of 1% Pluronic F-127 in PEM was added into the gasket wells and incubated at room temperature for 10 min to block the surface. Pluronic F-127 was then washed away twice with PEM20. The cover slip with the gasket was set up first on the microscope to find the surface, and then 6 μl of 2.5 nM eGFP–POK2$^{2083–2771}$ in PEM20 was added to the wells. Immediately after, 1.5 μl of hydrated GUVs was added to the wells. Images of GUVs were taken with our custom-built dual-colour TIRF setup (Schellhaus et al., 2017; Chugh et al., 2018; Wedler et al., 2022; Simmert et al., 2018) with the TIRF mirror positioned for highly inclined and laminated optical sheet (HILO) microscopy. The GUVs were unlabelled but could be detected by HILO microscopy when illuminated by reflections from fluorescent contaminations. GUVs with high PA content were difficult to make and only few GUVs formed. In these samples, there was more non-specific binding of POK2 clusters to the surface. The image focus plane was at the equator of the GUVs, several micrometres above the surface. Static diffraction patterns are visible in Fig. 7B from defocussed surface-bound POK2 clusters (Movie 1). Micromirrors directly below the objective that are used for illumination (Simmert et al., 2018) block a small part of the emission, causing wedges without emission on the left and right of the interference rings. Binding of eGFP–POK2$^{2083–2771}$ to GUVs was verified for at least three experimental repeats performed on different days.

### Acknowledgements
We acknowledge and appreciate the discussions with Yannic Lurz, Maria Kharlamova, Pantelis Livanos and Sabine Müller. We also thank Sabine Müller for sharing *Arabidopsis thaliana* cDNA.

### Competing interests
The authors declare no competing or financial interests.

### Author contributions
Conceptualization: S.Y.L., B.S.J.F., M.C., H.D., E.S.; Data curation: S.Y.L.; Formal analysis: S.Y.L., L.M., S.P., H.D.; Funding acquisition: E.S.; Investigation: S.Y.L., S.J., A.G.; Methodology: S.Y.L., H.D., E.S.; Project administration: S.Y.L., H.D., E.S.; Resources: E.S.; Supervision: E.S.; Validation: S.Y.L.; Visualization: S.Y.L., H.D.; Writing – original draft: S.Y.L., H.D.; Writing – review & editing: S.Y.L., H.D., E.S.

### Funding
S.Y.L. acknowledges financial support from and is a member of the International Max Planck Research School 'From Molecules to Organisms' at the Max Planck Institute for Biology, Tübingen and the University of Tübingen (Eberhard Karls Universität Tübingen). H.D. acknowledges funding by the Deutsche Forschungsgemeinschaft – project number 417890911. S.P. acknowledges funding by the Deutsche Forschungsgemeinschaft – project number 527934150. This work was supported by the Deutsche Forschungsgemeinschaft as part of the Collaborative Research Centre 1101 subproject A04 (to E.S.) and the University of Tübingen. Open Access funding provided by Eberhard Karls Universität Tübingen. Deposited in PMC for immediate release.

### Data and resource availability
All relevant data and details of resources can be found within the article and its supplementary information.

### First Person
This article has an associated First Person interview with the first author of the paper.

### Peer review history
The peer review history is available online at https://journals.biologists.com/jcs/lookup/doi/10.1242/jcs.263785.reviewer-comments.pdf

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
