## [Peer Review File · Journal of Cell Science]

The tail domain of the plant kinesin-12 POK2 is a versatile interaction hub

Shu Yao Leong, Laura Muras, Benedikt S. J. Fischer, Sehee Jang, Anastasia Gurskaya, Mayank Chugh, Serapion Pyrpasopoulos, Hauke Drechsler and Erik Schäffer
DOI: 10.1242/jcs.263785

Editor: Charlotte Kirchhelle

Review timeline

Original submission:	10 December 2024
Editorial decision:	4 February 2025
First revision received:	4 June 2025
Editorial decision:	1 July 2025
Second revision received:	7 August 2025
Accepted:	12 August 2025

Original submission

First decision letter

MS ID#: jcs.263785

MS TITLE: The plant kinesin-12 POK2 tail is a versatile connectivity hub

AUTHORS: Shu Yao Leong; Laura Muras; Benedikt Sebastian Jakob Fischer; Sehee Jang; Anastasia Gurskaya; Mayank Chugh; Serapion Pyrpasopoulos; Hauke Drechsler; Erik Schäffer
ARTICLE TYPE: Research Article

Dear Dr Schäffer,

Thank you for submitting your manuscript to Journal of Cell Science. We have now reached a decision on the above manuscript.

To see the reviewers' reports and a copy of this decision letter, please go to:

As you will see, the reviewers refer to the potential interest of the study, but they also indicate that further work would be necessary for publication. In particular, there were some concerns that the manuscript overstated the biological relevance of the observed interactions *in vitro*, and that *in vivo* data was required to underwrite some of the conclusions (specifically with respect to POK2 function at the CDZ vs the phragmoplast). There are also some methodological concerns, particularly regarding the quality of the lipid binding essays conducted in this study.

Reviewer 1

SUMMARY OF THE ADVANCE MADE IN THIS PAPER AND ITS POTENTIAL SIGNIFICANCE TO THE FIELD

In their manuscript, Leong et al. present an *in vitro* investigation into the binding of the C-terminal region of the POK2 kinesin with microtubules, the MAP65-3 protein, and lipids. Their findings demonstrate that the POK2 tail binds directly to microtubules and diffuses along them, supporting the hypothesis of a second microtubule binding site. Furthermore, the authors show that MAP65-3 enhances the microtubule binding activity of the POK2 tail. Additionally, they provide evidence that the POK2 tail binds directly to anionic lipids *in vitro*. The authors propose a model in which

POK2 recruitment to the PPB/CDZ occurs through sequential and cooperative interactions with cortical microtubules, MAP65-3, and the cortical membrane.

The *in vitro* data are robust, and the methods and results are well documented. The observed dual microtubule and lipid binding is intriguing. In particular, this work is one of the few studies, which addresses binding modes and complex formation of plant kinesins and MAPs *in vitro* and thus provides a valuable resource and novel insights. Overall, I believe the manuscript is of interest to the readership of JCS. However, I do not completely agree with the data interpretation. Without *in planta* analyses, the conclusions are not fully supported by the experiments. In my opinion, the current data provide a framework for future mechanistic studies rather than definitive conclusions. I thus recommend to avoid overstatements and to rephrase the respective parts of the manuscript accordingly.

Below are my detailed comments and suggestions:

SUGGESTIONS TO AUTHORS

The manuscript uses the term "higher plants," which is outdated. "Embryophytes" would be a more accurate term, as the PPB structure is specific to this group, with similar structures seen in some streptophytic algae.

The authors suggest cooperative binding of the POK2 C-terminus, with distinct POK2 clusters observed. Do these clusters indicate higher molecular weight complexes from direct binding? A size-exclusion chromatography (SEC) experiment followed by multi-angle light scattering (MALS) could address this.

Do the concentrations used (e.g., 50 nM) reflect *in vivo* POK2 concentrations? The punctate pattern observed at this concentration may resemble *in vivo* POK2 C terminal tail localization at the cortical membrane during interphase.

The authors mention that the last 100 amino acids of the POK2 C-terminus are critical for microtubule binding, likely due to electrostatic interactions with acidic tubulin tails. The authors state that POK2 C-terminus binding is sensitive to ionic strength. However, testing binding across a POK2 concentration gradient with varying salt conditions has not been systematically conducted. This could provide additional evidence for electrostatic contributions. In the same line: Does the removal of tubulin C-terminal tails affect POK2 and POK2-MAP65-3 complex binding to microtubules?

The study focuses on MAP65-3, but other MAP65 isoforms (e.g., MAP65-1, MAP65-4) are also involved in PPB function. There is some ambiguity in the manuscript regarding the role of MAP65-3 in CDZ targeting. If MAP65-3 is unlikely to be a key factor in POK2 recruitment to the CDZ, this section should be revised.

While lipid binding by POK2 is demonstrated, its relevance for CDZ localization is unclear. The authors should emphasize the need for further *in planta* studies to validate the role of lipid binding in POK2 recruitment to the CDZ. Additionally, they mention that MAP65-3 deficiency does not alter POK2 cortical localization, suggesting other cortical elements may be involved.

Is it known whether the POK2 C-terminus can bind TAN, as seen with POK1? Since TAN remains at the CDZ during cell division, it could be a more likely candidate for POK2 recruitment than MAP65-3. Vice versa, does POK1 interact with MAP65 proteins? The C-terminal tails of both, POK1 and POK2, localize to distinct clusters reminiscent of membrane microdomains in interphase cells (e.g., in *N. benthamiana*). This pattern is reminiscent of POK2 C-term (0283-2771) clusters that form by self-organization *in vitro*. Microtubule recruitment, however, is only observed upon coexpression with MAP65 proteins (in the case of POK2, or upon coexpression with IQDs (for both, POK1 and POK2) *in planta*). Why does the C-terminal tail bind microtubules *in vitro* with high affinity but not *in vivo* (at least in interphase cells)?

The use of an 8x His tag on all analyzed proteins could potentially enhance microtubule binding, as shown for EB1 in a previous study (DOI: 10.1074/jbc.M109.013466). The potential effect of the His tag on binding should be considered in the data interpretation. Perhaps this effect is stronger when the His-tag is positioned close to basic residues inside the protein of interest? Could this explain the lack of microtubule targeting of the POK1/2 C-terminal tails in the transient assays in *N. benthamiana*?

Please clarify the terminology around cardiolipins, as they are primarily associated with bacterial membranes or mitochondria, not plant membranes. In dividing *Arabidopsis* cells, PA is primarily found at the cell plate (<https://doi.org/10.3389/fpls.2019.00419>). Would this mean that lipid binding might be even more relevant for midzone confinement of POK2 rather than its CDZ

retention (or even recruitment in the absence of a PPB, as reported in trm678 and iqd678 mutants?). In line with this possibility, the motor domain of POK2 alone shows less confined midzone localization when compared to full length POK2.

Taking the possible role of lipids, microtubules, and MAP65-3 at the phragmoplast midzone/cell plate nexus into account the conclusions about CDZ localization are speculative without in planta validation. The introduction emphasizes dissecting the differences between POK2 pools at the CDZ and phragmoplast, but the study focuses primarily on MAP65-3, which is more relevant to the phragmoplast. Given these indications it would be helpful to emphasize that future work is needed to address the relevance of these in vitro findings to POK2 dynamics at the CDZ and phragmoplast in live cells.

Lastly, the authors mention GSK and cyclin-dependent kinases as potential regulators. Why are MPKs, which also are proline-guided S/T kinases sharing the same minimal motif, not discussed in this context? Did the authors check for presence of kinase docking sites for the three classes of proline-guided kinases?

Line 266: have been described (not describes)

Reviewer 2

SUMMARY OF THE ADVANCE MADE IN THIS PAPER AND ITS POTENTIAL SIGNIFICANCE TO THE FIELD

This manuscript by Leong and colleagues focuses on the interactions of POK2, uncovering potential functions of the C-terminal tail in microtubule and plasma membrane targeting and MAP65-3 binding. While it had been previously noted that POK2 could bind microtubules in the absence of the kinesin domain, and also binds to MAP65-3, this study has narrowed down the region in POK2 responsible for these interactions. Move novel is the finding that the C-terminal tail of POK2 can bind to anionic lipids. This is an interesting observation which may help explain retention of POK2 at the CDZ during mitosis. The manuscript rightly identifies that there will need to be significant follow up work to identify the mechanism, although a potential model is presented in the discussion and the final figure. How many kinesins have the potential to bind lipids is an interest question.

While I am not an expert in many of the methods used, the experiments seem thoroughly done and the conclusions reasonable, however a significant limitation of the study is that it relies solely on in vitro reconstitution experiments. The manuscript is very well written with sufficient explanation and interpretation in most places so as to make the study accessible to a non-specialist (exceptions as noted below).

SUGGESTIONS TO AUTHORS

I have a few points and a number of minor suggestions:

Line 187: How would this hypothesis that stationary events are not linked to microtubule binding fit with the events that switched between stationary and diffusive movement?

In the discussion, some speculation on how POK1 and POK2 can be (partially) functionally redundant might be helpful, given the POK1 C-terminal end is hypothesised not to mediate similar interactions.

What are the relative binding strengths of the C-terminal microtubule binding site and the ATP-dependent motor domain?

Line 456/478: how were 'greenest' and 'pinkest' determined?

Line 512 and Figure 1B: Bleed through of eGFP signal into rhodamine channel. How can you determine what is tubulin and what is bleed through in the second row? Would it be better to subtract the GFP signal from the rhodamine channel? Having a composite image which purports to

show overlap for all GFP signal due to bleed through in the rhodamine channel is perhaps not entirely useful and could be misinterpreted if the paper is not read carefully.

Line 563: How were bleaching and dissociation distinguished? It would be helpful to spell this out in the methods.

Figure 4B and 4D: There seems to be quite high mCherry signal for the 0 nM control. A comment on this in the text would be helpful.

Figure 7B: Last column of images, there seems to be a very similar pattern repeated multiple times. Is this an artifact?

Blot transparency panel D: the second gel image is rather pixelated. Can you please provide a higher resolution image.

Minor edits:

Line 368: 'microtubule binding domain' rather than 'microtubule domain'?

Line 364 and 380: Refers to Fig 8 E-I, however panels on Fig 8 stop at F.

Line 451: 'passed' rather than 'pass'?

Line 494: 'were' instead of 'was'?

Figure 1C: Units for the Y-axis?

Figure S5: 'long term' rather than 'long time' in the figure legend?

Reviewer 3

SUMMARY OF THE ADVANCE MADE IN THIS PAPER AND ITS POTENTIAL SIGNIFICANCE TO THE FIELD

Previous reports showed that the phragmoplast orienting kinesin 2 (POK2) was a weak motor by examining the microtubule-binding motor domain segment of the protein. In this manuscript, the authors show that the C-terminal fragment of POK2 binds microtubules, the microtubule associated protein MAP65-3, and various lipid *in vitro*. First, the C-terminal region comprised of amino acids 2083-2771 were used, followed by further truncations (to identify additional/supplementary microtubule binding sites) including amino acids 2083-2440, 2510-2771, and 2510-2677. While it was easy to follow based on the color coding of the constructs used within the manuscript and throughout the figures, I think it is unlikely to be preserved in the final publication. The authors are encouraged to find another way to clarify the fragments selected. Fascinatingly, MAP65-3 apparently enhances POK2 tail 2083-2771 binding to microtubules, but Figure 5 shows that increased binding occurs with a smaller fragment containing 2510-2771. Further, MAP65-3-POK2 tail binding appeared to reduce diffusive movement on microtubules. Some discussion about MAP65-3 binding enhancing POK2 binding to the CDS seemed inappropriate, considering MAP65-3 is primarily at the phragmoplast midline, and only transiently at the CDS. Lipid binding was demonstrated, but the data appeared unreplicated and unquantified. It was further unclear how this binding related to MAP65-3 or microtubule binding.

SUGGESTIONS TO AUTHORS

"The plant kinesin-12 POK2 tail is a versatile connectivity hub" is an overstatement, please rephrase to reflect the results of this entirely *in vitro* study focused on MAP65-3, microtubule, and lipid binding.

Figure 2A: the middle panel had no apparent microtubules. It was unclear what was meant by "2-3 technical replicates" in the figure legend. Please explain fully. Finally regarding Figure 2, it was

unclear why so few concentrations were tested for microtubule binding, especially for the 2510-2771 fragment. This seems a relevant concern, considering later figures that use smaller amounts of this fragment with MAP65-3.

POK2 and microtubule binding curves have only been calculated by fluorescence intensities. Another independent way to confirm this experiment would be microtubule co-sedimentation. Authors should have sufficient protein to test binding using this independent method. Authors themselves mention on L186 "Hence a majority if not all static binding events observed might reflect non-specific background binding of the protein rather than actual microtubule association". It was not clear what this meant, especially in context with Figure 6. Do the authors sometimes discount static binding, but analyze it in other scenarios? If this is the case, the authors should explain why.

It looked like lipid overlay assays were done once (?) from materials and methods and Figure 7. Additionally, GUV-POK2 tail binding was not only unquantified, but it was unclear why there was so much variability between images shown in Figure 7.

First revision

Author response to reviewers' comments

Reviewer 1:

SUMMARY OF THE ADVANCE MADE IN THIS PAPER AND ITS POTENTIAL SIGNIFICANCE TO THE FIELD

In their manuscript, Leong et al. present an in vitro investigation into the binding of the C-terminal region of the POK2 kinesin with microtubules, the MAP65-3 protein, and lipids. Their findings demonstrate that the POK2 tail binds directly to microtubules and diffuses along them, supporting the hypothesis of a second microtubule binding site. Furthermore, the authors show that MAP65-3 enhances the microtubule binding activity of the POK2 tail. Additionally, they provide evidence that the POK2 tail binds directly to anionic lipids in vitro. The authors propose a model in which POK2 recruitment to the PPB/CDZ occurs through sequential and cooperative interactions with cortical microtubules, MAP65-3, and the cortical membrane. The in vitro data are robust, and the methods and results are well documented. The observed dual microtubule and lipid binding is intriguing. In particular, this work is one of the few studies, which addresses binding modes and complex formation of plant kinesins and MAPs in vitro and thus provides a valuable resource and novel insights. Overall, I believe the manuscript is of interest to the readership of JCS. However, I do not completely agree with the data interpretation. Without in planta analyses, the conclusions are not fully supported by the experiments. In my opinion, the current data provide a framework for future mechanistic studies rather than definitive conclusions. I thus recommend to avoid overstatements and to rephrase the respective parts of the manuscript accordingly. Below are my detailed comments and suggestions:

SUGGESTIONS TO AUTHORS

[1] The manuscript uses the term "higher plants," which is outdated. "Embryophytes" would be a more accurate term, as the PPB structure is specific to this group, with similar structures seen in some streptophytic algae.

We have changed the respective terms accordingly.

[2] The authors suggest cooperative binding of the POK2 C-terminus, with distinct POK2 clusters observed. Do these clusters indicate higher molecular weight complexes from direct binding? A size-exclusion chromatography (SEC) experiment followed by multi-angle light scattering (MALS) could address this.

We agree with the reviewer that further knowledge of the POK2 C-terminus oligomerisation

status in solution would be helpful in interpreting our single-molecule TIRF microscopy data. However, our current purification protocol does not yield sufficient protein (total and concentration) for quantitative analyses like SEC-MALS or analytical ultracentrifugation (AUC). It was impossible to concentrate the POK2 C-terminus after purification, as the protein precipitated onto spin concentrator membranes, from which it could not be recovered. We now added native PAGE gels for a semiquantitative estimation of the oligomerisation states that the POK2 C-terminus has in solution (see Supplementary Figure 1C). These gels show that the constructs eGFP-POK2²⁰⁸³⁻²⁷⁷¹ and eGFP-POK2²⁰⁸³⁻²⁴⁴⁰ are mainly dimeric in solution under Blue Native- PAGE conditions. Our data suggests that deletion of the most C-terminal sequence of POK2 containing the second microtubule binding interface does not affect the tail oligomerisation state. However, deletion of amino acids 2083-2510, predicted to contain an extended coiled-coil stretch, completely abolished multimerization as POK2 constructs 2510-2771 and 2510-2677 were only monomeric in solution. Thus, we conclude that multimerisation of the POK2 C-terminal fragment creates multivalency that is required to shift its microtubule affinity from the micromolar range (i.e., POK2²⁵¹⁰⁻²⁷⁷¹, Figure 2) to the lower nanomolar range (i.e., POK2²⁰⁸³⁻²⁷⁷¹, Figure 1). We report these new findings in lines 145-147 and 169-170 and discuss them in lines 149-157 and 484-488.

[3] Do the concentrations used (e.g., 50 nM) reflect *in vivo* POK2 concentrations? The punctate pattern observed at this concentration may resemble *in vivo* POK2 C terminal tail localization at the cortical membrane during interphase.

To our knowledge, there is currently no published information on the *in vivo* concentrations of POK2. We contacted Sabine Müller, the leading expert in the plant kinesin-12 field, but she could also not provide any estimates for the POK2 concentration *in vivo*. We attempted to estimate it based on the relative abundance of POK2 and RBCL derived from the PaxDb database (Wang *et al.*, *Mol Cell Proteomics*, 2012). In *A. thaliana* the relative concentrations of POK2 and RBCL were 0.872 ppm and 41513 ppm, respectively (whole organism (integrated), 76% coverage). Assuming a RBCL copy number per cell of $3191 \cdot 10^6$ and a leaf cell volume of $0.03 \cdot 10^{-9}$ l (Heinemann *et al.*, *Plant Physiol*, 2021), we calculated the POK2 *in vivo* concentration to be ~4 nM. This value is likely an underestimation given that all kinesins in humans that drive spindle assembly or control microtubule dynamics have estimated concentrations between 20 and 400 nM according to the Open Cell database (Cho *et al.*, *Science*, 2022) (e.g., KIF11 - 350nM, KIF15 - 87nM, KIFC1 - 120nM, KIF2C - 260nM, KIF18A - 19nM, KIF18B - 28nM). Thus, the 50 nM POK2 tail fragment in our *in vitro* assay could very well reflect physiological conditions.

Still, we would like to refrain from any speculations, whether oligomeric POK2²⁰⁸³⁻²⁷⁷¹ particles observed *in vitro* might correspond to cortical punctae in interphase cells that occur, when the same construct is expressed *in vivo* (see Hermann *et al.*, *EMBO Rep*, 2018). The nature and physiological relevance of these punctae is unclear. The authors do not provide any expression level analyses for their different constructs and expression systems. Expression from a CaMV 35S promoter will likely result in overexpression of the respective protein such that punctae may be unspecific protein aggregations caused by overexpression. In line with this, their subcellular localisation is inconsistent in different expression systems. In *A. thaliana* root cells, punctae seem to co-localise with interphase microtubules (their Figure 3B). At the same time, this localisation appears not to be the case in *A. thaliana* protoplasts (their Figure EV2G), *N. benthamiana* leaf cells (their Figure 6G), or in root cells during their oryzalin experiment (their Figure EV1F). Punctae were also absent in interphase cells expressing the full-length constructs (see their Figures 2C, 5A, EV1C-E), suggesting that punctae formation is not an intrinsic trait of the full-length protein. See also our response to comment [9] for further considerations on *in vitro* clusters and *in vivo* punctae. We have addressed this in lines 149-157.

[4] The authors mention that the last 100 amino acids of the POK2 C-terminus are critical for microtubule binding, likely due to electrostatic interactions with acidic tubulin tails. The authors

state that POK2 C-terminus binding is sensitive to ionic strength. However, testing binding across a POK2 concentration gradient with varying salt conditions has not been systematically conducted. This could provide additional evidence for electrostatic contributions.

Figure 6A and B show that the landing-rate of eGFP-POK2²⁰⁸³⁻²⁷⁷¹ significantly reduces from 70 to 190 mM ionic strength, thereby supporting our conclusion that POK2 and tubulin bind through electrostatic interactions at the single molecule level. The reviewer, though, is right that our previous manuscript was missing a bulk experiment formally showing that POK2 interacts via electrostatic interactions with the microtubule lattice. Varying POK2 and salt concentration, however, may not fully answer this question, as increasing salt concentrations might interfere with protein stability or oligomerisation (see point [2]), thereby interfering with its microtubule binding ability. Given the limited amount of protein available within a reasonable time frame, we decided to invest it into the - more conclusive - experiment requested by the reviewer in point 5 that supports the notion that electrostatic interactions are key.

[5] In the same line: Does the removal of tubulin C-terminal tails affect POK2 and POK2-MAP65-3 complex binding to microtubules?

We added data on the binding of POK2 and POK2/MAP65-3 to microtubules treated with the subtilisin protease, which removes the negatively charged C-termini (E-hooks) of tubulin on the surface of the microtubule lattice. This removal suppressed binding of the POK2 C-terminus to microtubules (Figure 2C, D and lines 181), supporting that the POK2-MT interaction is mediated via electrostatic interactions. We also now show that the interaction of MAP65-3 with the microtubule lattice depends on the presence of the tubulin E-hooks in Supplementary Figure 1D. Accordingly, also the POK2-MAP65-3 complex did not bind to subtilisin treated microtubules (Supplementary Figure 5 and lines 248-250).

We have updated the results and discussion accordingly.

[6] The study focuses on MAP65-3, but other MAP65 isoforms (e.g., MAP65-1, MAP65-4) are also involved in PPB function. There is some ambiguity in the manuscript regarding the role of MAP65-3 in CDZ targeting. If MAP65-3 is unlikely to be a key factor in POK2 recruitment to the CDZ, this section should be revised.

According to *Smertenko et al., Plant Cell, 2008*, MAP65 paralogues 1/2, 3, 4, 5, and 6 localise to the PPB. MAP65-1, 3, and 5 were shown to interact with POK2 and, thus, could be responsible for targeting the motor's tail to the CDZ. As paralogue 3 had the strongest interaction with POK2 (i.e., via two distinct interaction sites in the N- and C-terminus of POK2 (*Hermann et al., EMBO Rep, 2018*)) and only the knockout of paralogue 3 resulted in a significant phragmoplast phenotype (*Müller et al., Curr Biol, 2004; Caillaud et al., Plant Cell, 2008; Ho et al., Plant Cell, 2012*), we considered it the most relevant paralogue for POK2 targeting. To what extent paralogues 1 and 5 also contribute to CDZ-targeting of POK2 needs to be determined by respective *in planta* experiments that are beyond the scope of the current manuscript.

We added new text in lines 345-352 to clarify the role of various isoforms.

[7] While lipid binding by POK2 is demonstrated, its relevance for CDZ localization is unclear. The authors should emphasize the need for further *in planta* studies to validate the role of lipid binding in POK2 recruitment to the CDZ. Additionally, they mention that MAP65-3 deficiency does not alter POK2 cortical localization, suggesting other cortical elements may be involved.

Describing cellular processes on the molecular level in general requires consistent *in vivo* data and *in vitro* insights. As our manuscript provides *in vitro* data on the protein level, we provide predictions on how we expect the described molecular behaviour will affect and/or explain

cellular processes *in vivo*. Thus, we consider our final model a working hypothesis that warrants further work - especially *in vivo* - to test it.

We have now emphasized the hypothetical nature of our model by adding respective statements in lines 347, 360, 372 and 392. A new *in vivo* study from Sabine Müller, published last week (Livanos *et al.*, *Nat Commun*, 2025), largely supports our main hypotheses.

According to our hypothesis, MAP65-3 or its paralogues (see [6]) are not a CDZ-targeting factor *per se*, but are required to (transiently) enrich and retain POK2 at the microtubule plus-ends, which in turn enables the efficient transfer of POK2 onto the plasma membrane. In the first version of this manuscript, we stated that there must be another mechanism that restricts membrane-bound POK2 to the CDZ - either by interaction with another CDZ component (as discussed and rejected for the CDZ component IQD8) or by the continuous focusing by the motors acting on peripheral phragmoplast microtubules.

We tried to make this point clearer in the updated manuscript.

[8] Is it known whether the POK2 C-terminus can bind TAN, as seen with POK1? Since TAN remains at the CDZ during cell division, it could be a more likely candidate for POK2 recruitment than MAP65-3. Vice versa, does POK1 interact with MAP65 proteins?

Currently, we do not think that TAN can be a CDZ-targeting for POK2 for the following reasons:

1. To our knowledge, there is no published evidence that TAN and POK2 physically interact. Müller *et al.*, *Curr Biol*, 2006 suggest that this MIGHT be the case - based on their observations that POK1 and TAN interact in a yeast 2 hybrid (Y2H) assay and that POK1 and POK2 are functionally redundant. This assumption, however, was not backed by a POK2/TAN Y2H or any other protein interaction assay in this publication.
2. CDZ-localisation of TAN itself depends on POK1/POK2 activity/presence as shown by Walker *et al.*, *Curr Biol*, 2007, and Lipka *et al.*, *Plant Cell*, 2014, so that it cannot act as a cortical anchor for those motors on its own.

To our knowledge, there is also no evidence in the literature that POK1 can interact with any of the MAP65 paralogues.

We now point out that it is unknown whether POK2 interacts with TAN or POK1 with MAP65 paralogues (see lines 399-408, 492).

[9] The C-terminal tails of both, POK1 and POK2, localize to distinct clusters reminiscent of membrane microdomains in interphase cells (e.g., in *N. benthamiana*). This pattern is reminiscent of POK2 C-term (0283-2771) clusters that form by self-organization *in vitro*. Microtubule recruitment, however, is only observed upon coexpression with MAP65 proteins (in the case of POK2, or upon coexpression with IQDs (for both, POK1 and POK2) in planta. Why does the C-terminal tail bind microtubules *in vitro* with high affinity but not *in vivo* (at least in interphase cells)?

Our new data in Figure S1C suggests that POK2²⁰⁸³⁻²⁷⁷¹ can form up to tetramers in solution *in vitro*. The clusters that form on the microtubule, appear to contain higher copy numbers based on their intensity. Cluster formation *must depend* on the POK2²⁰⁸³⁻²⁷⁷¹ interaction with the microtubule, as we cannot observe unspecific background-binding of high-intensity clusters to the cover slip. We also do not see any clusters upon dilution below about 15 nM. Thus, microtubules might concentrate POK2²⁰⁸³⁻²⁷⁷¹ locally, allowing the formation of higher-order oligomers, or microtubule binding might lead to a conformational change of POK2²⁰⁸³⁻²⁷⁷¹ that favours the formation of larger oligomers.

We already discussed in point [3] that the nature of those cortical POK2²⁰⁸³⁻²⁷⁷¹ punctae in interphase cells is at least ambiguous. In most cases, they appear to form in the absence of microtubules, suggesting that higher clusters *in vitro* and cortical punctate *in vivo* are not the same. In the case where high intensity clusters *in vitro* and interphase punctate *in vivo* might be the same, our work hints at possible explanations why POK2²⁰⁸³⁻²⁷⁷¹ *in vivo* only binds microtubules in the presence of MAP65-3 in interphase:

Cells need to strictly control their microtubule bundling activity in space and time to prevent potentially detrimental distortions of their microtubule cytoskeleton. Hence, especially in interphase, many mitotic kinesins are either not expressed, auto-inhibited, or otherwise inactivated by the presence or absence of posttranslational modification. In line with this, the C-terminal end of POK2 contains CDK and GSK3 phosphorylation consensus sites (see also point [13] suggesting POK2 is phosphorylated in a cell-cycle dependent manner. Therefore, we speculate that the second microtubule binding domain of POK2 may be regulated by phosphorylation in interphase, explaining why POK2²⁰⁸³⁻²⁷⁷¹ does not bind microtubules when overexpressed in interphase. Our data also shows that the interaction with MAP65-3 drastically increases the microtubule affinity of POK2²⁰⁸³⁻²⁷⁷¹, explaining why this construct then reroutes to microtubules in interphase cells only when MAP65-3 is co-expressed. This interpretation remains speculative and needs to be tested by future *in vivo* work using non-phosphorylatable POK2 mutants.

We now speculate in the discussion that phosphorylation may regulate POK2 binding in a cell-cycle dependent manner (see lines 513-523).

[10] The use of an 8x His tag on all analyzed proteins could potentially enhance microtubule binding, as shown for EB1 in a previous study (DOI: 10.1074/jbc.M109.013466). The potential effect of the His tag on binding should be considered in the data interpretation. Perhaps this effect is stronger when the His-tag is positioned close to basic residues inside the protein of interest? Could this explain the lack of microtubule targeting of the POK1/2 C-terminal tails in the transient assays in *N. benthamiana*?

We thank the reviewer for pointing out a potential contribution of the His-Tag to microtubule binding. While the positive effect of the His-tag on EB1 microtubule binding in the cited study (Zhu *et al.*, *JBC*, 2009) is more likely to be caused by the His-tag interfering with EB1 autoinhibition (see Kanaba *et al.*, *BBA*, 2013), a recent study (Inaba *et al.*, *Sci Adv*, 2022) clearly shows that the His-tag can mediate microtubule binding of tetrameric fluorescent proteins, though in the μM -range, only. Hence, we consider the contribution of the His-tag to the overall POK2-tail microtubule binding activity, which occurs already in the low nanomolar range in our *in vitro* assay - to be negligible. In line with that, the microtubule affinities of our constructs differ by more than an order of magnitude, although they all carry a His-tag. We address this caveat now in lines 187-188.

Accordingly, we also don't think that the 'missing' His-tag could explain the lack of microtubule binding of the POK2²⁰⁸³⁻²⁷⁷¹ construct when overexpressed *in vivo*. Please also refer to our alternative explanations for this observation under points [3] and [9].

[11] Please clarify the terminology around cardiolipins, as they are primarily associated with bacterial membranes or mitochondria, not plant membranes. In dividing *Arabidopsis* cells, PA is primarily found at the cell plate (<https://doi.org/10.3389/fpls.2019.00419>). Would this mean that lipid binding might be even more relevant for midzone confinement of POK2 rather than its CDZ retention (or even recruitment in the absence of a PPB, as reported in trm678 and iqd678 mutants?). In line with this possibility, the motor domain of POK2 alone shows less confined midzone localization when compared to full length POK2.

We thank the reviewer for bringing this up. Indeed, we have not been explicit about

cardiolipin's absence from plant plasma membranes. We have added clarification in line 391-392.

The reference the reviewer kindly provided refers to *Caillaud, 2019 (Front. Plant Sci.)* which refers to *Platre et al., 2018 (Dev. Cell)*, specifically Video S1 of this publication where PA, visualised by the endogenously integrated biosensor mCIT-1xPASS, is observed to localise to the plasma membrane and becomes enriched in the cell plate during division. Indeed, this point could be relevant for the midzone confinement of the entire POK2 protein. However, lipid binding of the POK2 N-terminal domain, which is responsible for phragmoplast midzone recruitment, has not been assessed, and we are thus unable to comment on this. On the other hand, the POK2 tail when expressed in the *pok1/pok2* mutant background fails to localise to the midzone and is only found at the CDZ (*Herrmann et al., EMBO Rep., 2020*). Hence, the PA enrichment at the cell plate does not enhance POK2 tail recruitment.

[12] Taking the possible role of lipids, microtubules, and MAP65-3 at the phragmoplast midzone/cell plate nexus into account the conclusions about CDZ localization are speculative without in planta validation. The introduction emphasizes dissecting the differences between POK2 pools at the CDZ and phragmoplast, but the study focuses primarily on MAP65-3, which is more relevant to the phragmoplast. Given these indications it would be helpful to emphasize that future work is needed to address the relevance of these in vitro findings to POK2 dynamics at the CDZ and phragmoplast in live cells.

Please refer to our detailed answers to point [7] for general remarks on the validity and intention of our model as a testable working hypothesis. Also refer to point [6], explaining why this work focuses on MAP65-3.

[13] Lastly, the authors mention GSK and cyclin-dependent kinases as potential regulators. Why are MPKs, which also are proline-guided S/T kinases sharing the same minimal motif, not discussed in this context? Did the authors check for presence of kinase docking sites for the three classes of proline-guided kinases?

We proposed that CDKs and GSK-3 might regulate the second microtubule binding site in the C-terminus of POK2 as (i) we could identify overlapping consensus sites of these two kinases within the second MBD and (ii) the same kinases control the secondary MBD in other kinesins as well (*Drechsler et al., JCS, 2015*). As suggested by the reviewer, we reinvestigated the secondary MBD, asking whether the putative phosphorylation sites in question (i.e., site A: MPS^PP and site B: SQS^PP) could be modified by other proline-guided kinases. MAPKs in general recognise a PXS/TP consensus (*Gonzales et al., JBC, 1991*) that is not given in both sites. The first proline might also be substituted with a hydrophobic amino acid (*Miller and Turk, Biochemical Sciences, 2018*), which then might qualify site A as a MAPK target. However, site A contains a proline preceding the serine (PSP), which had been shown to largely abolish MAPK-dependent phosphorylation at this site (*Gonzales et al., JBC, 1991*). We hence do not see any indication that MAPKs might be involved in the regulation of POK2's secondary MBD.

We expanded the discussion, pointing out that these kinases likely do not interact with the POK2 C-terminus (see lines 516-523).

Line 266: have been described (not describes)

We corrected the text.

Reviewer 2:

SUMMARY OF THE ADVANCE MADE IN THIS PAPER AND ITS POTENTIAL SIGNIFICANCE TO THE FIELD

This manuscript by Leong and colleagues focuses on the interactions of POK2, uncovering

potential functions of the C-terminal tail in microtubule and plasma membrane targeting and MAP65-3 binding. While it had been previously noted that POK2 could bind microtubules in the absence of the kinesin domain, and also binds to MAP65-3, this study has narrowed down the region in POK2 responsible for these interactions. More novel is the finding that the C-terminal tail of POK2 can bind to anionic lipids. This is an interesting observation which may help explain retention of POK2 at the CDZ during mitosis. The manuscript rightly identifies that there will need to be significant follow up work to identify the mechanism, although a potential model is presented in the discussion and the final figure. How many kinesins have the potential to bind lipids is an interesting question.

While I am not an expert in many of the methods used, the experiments seem thoroughly done and the conclusions reasonable, however a significant limitation of the study is that it relies solely on *in vitro* reconstitution experiments. The manuscript is very well written with sufficient explanation and interpretation in most places so as to make the study accessible to a non-specialist (exceptions as noted below).

SUGGESTIONS TO AUTHORS

I have a few points and a number of minor suggestions:

[14] Line 187: How would this hypothesis that stationary events are not linked to microtubule binding fit with the events that switched between stationary and diffusive movement?

Although we passivate our TIRF-assay glass chambers with a non-ionic copolymer surfactant (i.e., Pluronic F-127), some residual non-specific background binding of POK2 particles to the passivated glass coverslips still occurs. The microtubule diameter of ~25 nm is considerably smaller than the (diffraction-limited) resolution of our optical setup. Single-molecules have a diameter of about 500 nm in the images corresponding to about 5 pixels (105 nm per pixel). Thus, we cannot resolve and discriminate between particles that bind directly to the microtubule and those that bind to the surface just next to the microtubule. While this apparent broadening of objects due to diffraction has no consequences for the quantification of dynamic particles (surface-bound particles do not move at all, and thus motile particles moving along a line and not randomly in two dimensions must be microtubule-associated), it complicates the analysis of static particles. The number of static particles must be compared to a control area far from the microtubule to estimate the fraction of non-specific, surface-bound particles in the microtubule particle data. In our case, the statistics of stationary particles 'on' the microtubule are indifferent to those of background control areas (see Figure S3A, B). Hence, we stated in the first version of the manuscript that '*a majority, if not all, static binding events [...] reflect [...] non-specific background binding*' line 207. In case of rarely observed switching events, we are likely observing events, in which surface-bound particles convert to microtubule-bound ones and vice versa (see scheme). We mention this possibility now in the text lines 216-218.

Scheme showing the (approximate) relative sizes in our *in vitro* reconstitution assay compared to

the pixel size of our optical setup. Please note that the diameter of a microtubule is approximately only a fifth of the pixel size (for the setup with the larger pixel size of 149 nm/pixel - see methods), so that a lot of unspecific background signal is integrated into this pixel as well.

[15] In the discussion, some speculation on how POK1 and POK2 can be (partially) functionally redundant might be helpful, given the POK1 C-terminal end is hypothesised not to mediate similar interactions.

We have expanded the discussion and speculate that while POK2 incorporates all functions required for CDZ targeting and phragmoplast orientation, the second microtubule interaction domain and the CDZ targeting activity may have been outsourced to other proteins in the case of POK1. See lines 488- 502.

[16] What are the relative binding strengths of the C-terminal microtubule binding site and the ATP-dependent motor domain?

We estimated the binding strength of the two microtubule binding sites when subjected to forces. The motor domain could generate only about 0.3 pN and then likely switches to a diffusive state. Based on the diffusion coefficient of the C-terminal domain, we estimated the friction force between this domain and the microtubule if the domain is being pulled at a certain speed over the microtubule. For the C-terminal domain alone, the expected friction force is less than the motor-generated force. However, if the C-terminal domain is in complex with MAP65-3, friction forces exceed motor forces, suggesting that MAP65-3 constrains the motor. In terms of binding affinity, both bind with nM affinities, with the motor domain having a higher affinity based on the number of interactions under single-molecule conditions (not quantified).

We attempted to make the discussion clearer on that point in lines 451-452.

[17] Line 456/478: how were 'greenest' and 'pinkest' determined?

We apologize for the imprecise language. As these terms should refer to the peak fractions, we changed the text accordingly, lines 551 and 578.

[18] Line 512 and Figure 1B: Bleed through of eGFP signal into rhodamine channel. How can you determine what is tubulin and what is bleed through in the second row? Would it be better to subtract the GFP signal from the rhodamine channel? Having a composite image which purports to show overlap for all GFP signal due to bleed through in the rhodamine channel is perhaps not entirely useful and could be misinterpreted if the paper is not read carefully.

At low POK2 concentrations, microtubules can be clearly distinguished from the GFP signal. At higher POK2 concentrations, microtubules are more or less completely decorated, such that a line pattern is visible in the GFP channel, and the microtubule signal would not be necessary. In addition, we can observe microtubules simultaneously in interference reflection microscopy (IRM), similar to the data in Figure 4, but not shown in Figure 1. If we needed the fluorescent tubulin signal for quantification, we could, as suggested by the reviewer, account for the bleed-through. However, this quantification was unnecessary here.

[19] Line 563: How were bleaching and dissociation distinguished? It would be helpful to spell this out in the methods.

We cannot discriminate between a lattice dissociation event and the bleaching of the last lit GFP in our TIRF assay. Therefore, these events are not considered in the bleaching probability analysis. If a multimer bleaches in several steps, we can quantify the duration of these bleaching events as long as at least one fluorophore still emits light. We now describe this procedure in

more detail in lines 688-690.

[20] Figure 4B and 4D: There seems to be quite high mCherry signal for the 0 nM control. A comment on this in the text would be helpful.

The logarithmic scale of the signal may be deceptive. In essence, the signal of the 0 nM control is the background noise of the camera and some background fluorescence due to autofluorescent contamination in the buffer. On a linear scale, these control signals would be close to zero with a background subtraction. In Fig. 4B at 0.1 nM, the signal is already a factor 10 higher. We now point out that signals are plotted on a logarithmic scale and that the control signals correspond to the camera noise and background fluorescence.

[21] Figure 7B: Last column of images, there seems to be a very similar pattern repeated multiple times. Is this an artifact?

Yes, indeed, this is an artefact, a combination of non-specific surface binding of POK2 and an asymmetric point-spread function specific to our custom-built TIRF microscope. Pure PA GUVs were difficult to make and only few GUVs formed. In these samples, there was more non-specific binding of POK2 clusters to the surface. If these clusters were imaged on the surface, they would be diffraction-limited points. However, the microscope focus plane was at the equator of the GUVs, i.e., several microns above the surface. The peculiar patterns are the diffraction pattern from the defocused surface-bound POK2 clusters. These patterns did not consist of continuous interference fringes because micromirrors that are directly below the objective used for illumination (Simmert *et al.*, 2018) block a small part of the emission (wedges without emission on the left and right). The patterns are also static in the now-provided videos. GUV-bound POK2 diffused on the spherical surface of the GUVs and can be clearly distinguished in the videos.

Apart from providing the videos, we now point out this artefact in the methods section, lines 739-745 (linked to in the main text at line 310-312).

[22] Blot transparency panel D: the second gel image is rather pixelated. Can you please provide a higher resolution image.

We would like to apologise for the poor image quality. This may have happened during the down conversion of the PDF file during the submission. We now provide high-resolution images.

Minor edits:

Line 368: 'microtubule binding domain' rather than 'microtubule domain'?

Line 364 and 380: Refers to Fig 8 E-I, however panels on Fig 8 stop at F.

Line 451: 'passed' rather than 'pass'?

Line 494: 'were' instead of 'was'?

Figure 1C: Units for the Y-axis?

Figure S5: 'long term' rather than 'long time' in the figure legend?

We thank the reviewer for pointing out these oversights. We corrected them accordingly.

Reviewer 3:**SUMMARY OF THE ADVANCE MADE IN THIS PAPER AND ITS POTENTIAL SIGNIFICANCE TO THE FIELD**

Previous reports showed that the phragmoplast orienting kinesin 2 (POK2) was a weak motor by examining the microtubule-binding motor domain segment of the protein. In this manuscript, the authors show that the C-terminal fragment of POK2 binds microtubules, the microtubule associated protein MAP65-3, and various lipid *in vitro*. First, the C-terminal region comprised of amino acids 2083-2771 were used, followed by further truncations (to identify additional/supplementary microtubule binding sites) including amino acids 2083-2440, 2510-2771, and 2510-2677. While it was easy to follow based on the color coding of the constructs used within the manuscript and throughout the figures, I think it is unlikely to be preserved in the final publication. The authors are encouraged to find another way to clarify the fragments selected. Fascinatingly, MAP65-3 apparently enhances POK2 tail 2083-2771 binding to microtubules, but Figure 5 shows that increased binding occurs with a smaller fragment containing 2510-2771. Further, MAP65-3-POK2 tail binding appeared to reduce diffusive movement on microtubules. Some discussion about MAP65-3 binding enhancing POK2 binding to the CDS seemed inappropriate, considering MAP65-3 is primarily at the phragmoplast midline, and only transiently at the CDS. Lipid binding was demonstrated, but the data appeared unreplicated and unquantified. It was further unclear how this binding related to MAP65-3 or microtubule binding.

We thank the reviewer for pointing out the issue of colour coding our constructs in the text and we will be discussing with the editor whether publication with colour coding in text is possible. Nevertheless, we have written our manuscript and designed our figures keeping in mind that the colour coding might not be retained and have been as explicit and clear as possible when introducing the constructs, in order to maintain logical flow:

(1) We have introduced the shorter tail constructs as the tail “cut” in half (eGFP- POK2²⁰⁸³⁻²⁴⁴⁰, eGFP-POK2²⁵¹⁰⁻²⁷⁷¹) according to their coiled-coil predictions, and the eGFP-POK2²⁵¹⁰⁻²⁶⁷⁷ construct as the same second half but lacking the final unstructured region. We named the constructs with their amino acid residue numbers, so that the length and truncation points can be directly inferred.

(2) We introduced the cartoon schematic of the constructs in Figure 1A with colour coding and used this same reference in a smaller format (square icons) for the rest of the figures.

(3) Together with the colour coding, the full names of the constructs are explicitly written in the figures.

In the event that publication with colour coding in text is not possible, we believe that our intentional introduction and reference of the constructs will still allow the reader to follow the manuscript and figures smoothly.

SUGGESTIONS TO AUTHORS

[23] “The plant kinesin-12 POK2 tail is a versatile connectivity hub” is an overstatement, please rephrase to reflect the results of this entirely *in vitro* study focused on MAP65-3, microtubule, and lipid binding.

We have changed the manuscript title to better reflect our results.

[24] Figure 2A: the middle panel had no apparent microtubules. It was unclear what was meant by “2-3 technical replicates” in the figure legend. Please explain fully. Finally regarding Figure 2, it was unclear why so few concentrations were tested for microtubule binding, especially for the 2510-2771 fragment. This seems a relevant concern, considering later figures that use smaller amounts of this fragment with MAP65-3.

In Figure 2A, we chose three representative concentrations. For quantification in Figure 2B, we

measured between 5 and 7 concentrations. More than 7 concentrations (for POK2²⁵¹⁰⁻²⁷⁷¹) would not provide more information for the binding curve because they cover already two orders of magnitude in signal (note that the vertical axis is logarithmic and the lines are guides to the eye). The concentration range was based on the minimal concentration, for which microtubule binding was apparent in the images, and the maximal concentration that could be achieved with the respective stock concentration. While it is correct that Figure 2B does not provide binding data for lower, nanomolar concentrations, this lack does not affect our conclusions that POK2²⁵¹⁰⁻²⁷⁷¹ binds with lower affinity to microtubules compared to POK2²⁰⁸³⁻²⁷⁷¹ or that the constructs POK2²⁰⁸³⁻²⁴⁴⁰ and POK2²⁵¹⁰⁻²⁶⁷⁷ show hardly any microtubule binding activity even compared to the already low binding construct POK2²⁵¹⁰⁻²⁷⁷¹. The logarithmic scale may be deceiving.

We now plotted the data also on a linear scale and point out that the lines are guides to the eye. We use the standard definition of technical replicates (vs. biological replicates) by which we mean that we repeated the measurement on different days (but did not, for example, do a new purification of the proteins). We rephrased the figure caption that the data are averages from 2-3 repeated measurements and provide more details in the methods.

In the middle panel, the microtubule sample contained a lot of small microtubule fragments that obscured the longer microtubules. We chose another image.

Our conclusions in Figure 5 are not affected by not quantifying the binding amount in Figure 2 for the POK2²⁵¹⁰⁻²⁷⁷¹ fragment. Figures 5A and C show that the presence of 20nM MAP65-3 caused POK2²⁵¹⁰⁻²⁷⁷¹ at 20 nM to bind microtubules (Figure 5C *right boxplot*). Without MAP65-3, POK2²⁵¹⁰⁻²⁷⁷¹ (20 nM) did not bind (Figure 5C *left boxplot*). In other words, this figure comes with an internal control giving the baseline for microtubule binding of POK2²⁵¹⁰⁻²⁷⁷¹ at 20nM. Thus, it does not require low concentration data in Figure 2B for its interpretation.

The figure and legend for Figure 2 have been updated and should be more precise now.

[25] POK2 and microtubule binding curves have only been calculated by fluorescence intensities. Another independent way to confirm this experiment would be microtubule co-sedimentation. Authors should have sufficient protein to test binding using this independent method. Authors themselves mention on L186 "Hence a majority if not all static binding events observed might reflect non-specific background binding of the protein rather than actual microtubule association". It was not clear what this meant, especially in context with Figure 6. Do the authors sometimes discount static binding, but analyze it in other scenarios? If this is the case, the authors should explain why.

Our TIRF microscope setup has a diffraction-limited resolution with diameters of diffraction-limited objects of about 500 nm, corresponding to about 5 pixels on the camera (105 nm per pixel - also see response to point [14] of Reviewer #2). Hence, we cannot resolve and discriminate whether a fluorescent particle is on the microtubule (25 nm in diameter) or on the glass coverslip next to the microtubule. Since background binding in general is static, dynamic particles must be on the microtubule. Static particles, in contrast, could be bound to the microtubule or as background to the coverslip. So, binding statistics of microtubule scans must be compared to background scans to assess the number of bona fide microtubule binding events. We did this analysis in Supplementary Figure 3A, B, showing no significant difference in the occurrence of background and static microtubule binding events. Thus, we concluded that most, if not all, static binding events in microtubule line scans are background binding events and should not be attributed to microtubule binding.

We clarify this now upfront in lines 214-216.

For our fluorescence-based binding analysis, static particles contribute to the fluorescence along the microtubule. However, their contribution is removed by subtracting the local non-specific background binding activity next to the microtubule. Given that this assay registers the binding of single molecules to a single microtubule, it is far more sensitive and accurate than a bulk method like a microtubule co-sedimentation assay. Such assays may be affected by partial microtubule depolymerisation, unspecific protein precipitation in the pellet, non-homogeneous staining of gels, pipetting errors, and potential cross-contamination of the supernatant and pellet fractions.

Nevertheless, we tried to provide the microtubule co-sedimentation data requested. We first tried a classical microtubule co-sedimentation assay setup, in which protein and microtubules are mixed and then sedimented by high-speed centrifugation. As expected for a protein that can oligomerise and tends to assemble in clusters, the majority of POK2²⁰⁸³⁻²⁷⁷¹ already precipitates in the absence of microtubules during a pre-clearance step (see right panel A, *below*). Thus, when spinning down microtubules, it is unclear whether spun-down POK2 was microtubule-bound or in solution in the form of a cluster. We then tried to spin the microtubule POK2²⁰⁸³⁻²⁷⁷¹ through a 60% cushion to prevent unspecific protein precipitation. While this prevented unspecific precipitation, the majority of microtubule bound POK2²⁰⁸³⁻²⁷⁷¹ dissociated upon passage through the (POK2-free) cushion (see right panels B, *below*). Thus, the Western-blot signal in the pellet was only a fraction of the POK2²⁰⁸³⁻²⁷⁷¹ that initially bound to the microtubule. Taken together, microtubule-sedimentation assays were not suitable to describe POK2 tail microtubule binding quantitatively, which is why we did not include these experiments in the final version of the manuscript.

POK2 2083-2771-microtubule co-sedimentation assay

(A) Co-sedimentation of eGFP POK2 2083-2771 at the indicated concentrations (*upper panels*, anti-GFP Western- blot) with 3 μ M double-stabilised microtubules (*lower panels*, Ponceau staining). Panels on the right show the protein that pelleted out during the pre-spin. *I* - input, *SN* - supernatant, *P*- pellet.

(B) Co-sedimentation of eGFP POK2 2083-2771 at the indicated concentrations (*upper panels*, anti-GFP Western- blot) with 3 μ M double-stabilised microtubules (*lower panel*, Ponceau staining)

through a **60% glycerol cushion**. Panels on the right show the protein content of the corresponding cushions after centrifugation. *I* - input, *SN* - supernatant, *P* - pellet.

Methods

Microtubule-sedimentation without a cushion

eGFP-POK2 2083-2771 was pre-cleared at 80,000 g for 5 min (Beckman Airfuge, rotor A-110, polypropylene tube #344718). The supernatant of this precleared protein was mixed with 2 μ M double-stabilised microtubules in a PEM20-based reaction mix containing 3.33 μ M paclitaxel, 0.2 mg/ml casein, 1 mM ATP and 5 mM DTT to a final volume of 20 μ l. eGFP-POK22083-2771 was prediluted in its storage buffer to maintain a constant ionic strength of ~130 mN (as in Fig. 1B & C) across the concentrations tested. The protein-microtubule mix was gently mixed with a cut-tip to reduce shearing and binding was allowed for 15 min at 30°C. The mix was then centrifuged for 5 min at 80,000 g. The supernatant was very gently pipetted out for analysis. To avoid cross-contamination, the pellet was washed very gently once with 20 μ l buffer mix and then recovered in 20 μ l buffer mix. Samples were then analysed by SDS-PAGE on a 4-20% Mini-PROTEAN TGX Precast gel (Bio-rad, 4561096) and subsequent western blotting. eGFP-POK2 2083-2771 was visualised by a anti-GFP (Roche, 11814460001, at 1:2000) and anti-mouse secondary antibody (Optimised HRP reagent at 1:20 from Pierce Fast Western Kit, 35060). The blot membrane was further stained with Ponceau to visualise tubulin loading.

Microtubule-sedimentation through a cushion

eGFP-POK22083-2771 was pre-cleared at 80,000 g for 5 min (Beckman Airfuge, rotor A-110, polypropylene tube #344718) on top of a 60% glycerol cushion. Precleared protein was mixed with 3 μ M double-stabilised microtubules in a PEM20-based reaction mix containing 3.33 μ M paclitaxel, 0.2 mg/ml casein, 1 mM ATP and 5 mM DTT to a final volume of 20 μ l. eGFPPOK2 2083-2771 was prediluted in its storage buffer to maintain a constant ionic strength of ~130 mN (as in Fig. 1B & C) across the concentrations tested. The protein-microtubule mix was gently mixed with a cut-tip to reduce shearing and binding was allowed for 20 minutes at 30°C. The mix was then layered on top of a 50 μ l 60% glycerol cushion (same buffer as supernatant) and centrifuged for 5 min at 80,000 g. To avoid perturbation of the cushion, the very top 15 μ l of each supernatant were removed for analysis, while the rest was discarded along with the cushion. The pellet was recovered in 20 μ l buffer mix. Samples were then analysed by SDS-PAGE on a 4-20% Mini-PROTEAN TGX Precast gel (Bio-rad, 4561096) and subsequent western blotting. eGFP-POK2 2083-2771 was visualised by a anti-GFP (Roche, 11814460001, at 1:2000) and anti-mouse secondary antibody (Optimised HRP reagent at 1:20 from Pierce Fast Western Kit, 35060). The blot membrane was further stained with Ponceau to visualise tubulin loading.

[26] It looked like lipid overlay assays were done once (?) from materials and methods and Figure 7. Additionally, GUV-POK2 tail binding was not only unquantified, but it was unclear why there was so much variability between images shown in Figure 7.

We have now updated Figure 7A, with a graph showing the average dot blot densities of three independent experiments for each construct. While technical replicates are highly consistent qualitatively, the absolute density values vary due to the non-linear signal development by the HRP/ECL detection method. As these variations still exist to some extent after normalisation to an internal POK2 standard (i.e., 200 ng POK2), we decided to show the single experiments in Supplementary Figure 6, so that readers can judge themselves.

Due to the artificial lipid organisation on the membrane during the lipid overlay assays, we performed the GUV-binding experiments as an independent test whether binding still occurs in a more natural setting. Since GUVs were generated by swelling of rehydrated lipids from a PVA film, GUV preparations are expected to be inhomogeneous. In particular, GUVs with high CL or PA content had a low yield and high variability. Variability in GUV size does not affect the qualitative conclusion that POK2 binds lipid membranes of a GUV. Any quantitative statements on the POK2 binding affinity to GUVs would require a substantial amount of work that we think is beyond the scope of this paper. Since binding to the GUVs is less clear in static images, we have

now added Movie 1 showing time lapse recordings of those GUVs shown in Figure 7. The movie shows the dynamic nature of the POK2 tail diffusing on the surface of the GUV (see also point [21] of Reviewer #2).

Second decision letter

MS ID#: jcs.263785R1

MS TITLE: The plant kinesin-12 POK2 tail is a versatile interaction hub

AUTHORS: Shu Yao Leong; Laura Muras; Benedikt Sebastian Jakob Fischer; Sehee Jang; Anastasia Gurskaya; Mayank Chugh; Serapion Pyrpasopoulos; Hauke Drechsler; Erik Schäffer

ARTICLE TYPE: Research Article

Dear Dr Schäffer,

We have now reached a decision on the above manuscript.

To see the reviewers' reports and a copy of this decision letter, please go to:

As you will see, all of the reviewers (and myself) greatly appreciated the substantial revisions you have conducted, which have further improved the quality of this manuscript. One of the reviewers still had some minor concerns. I hope you are willing to address these in a final revision - I would like to stress I expect only minor modifications to the manuscript and no further experiments at this stage.

Reviewer 1

This is the revised version of a manuscript I previously reviewed. I thank the authors for their thorough responses and the additional experimental data, which convincingly support the main conclusions. Overall, the manuscript has significantly improved and - in my opinion - is suitable for publication in JCS. The study is also particularly timely, as it aligns with recent findings published in Nature Communications (Livanos et al, 2025), highlighting its relevance to current developments in the field. In my view, the work should not be delayed further.

Reviewer 2

Having looked at the revised version of the manuscript and read the extensive response to the reviewers, I feel the authors have done a thorough job of appropriately addressing the points I raised. In my opinion, the manuscript is now appropriate for acceptance.

Reviewer 3

SUMMARY OF THE ADVANCE MADE IN THIS PAPER AND ITS POTENTIAL SIGNIFICANCE TO THE FIELD

Revision

SUGGESTIONS TO AUTHORS

Figure 2A (middle panel) tubulin only channel looks very different from the other tubulin only channels in other parts of the figure and the rest of the manuscript. The authors are encouraged to find other images (?). This was possibly mentioned in the last review.

Similarly, in Figure 5, with the same construct (middle panel), why is the background so high for 20 nm protein?

The addition of subtilisin treatment to remove E-hooks was interesting. Was the spelling correct in the figures and manuscript?

Thank you for providing microtubule co-sedimentation experiments. The widely variable tubulin concentrations in the bottom panels were unexpected. Why is there so little tubulin in the pellet fraction in 100 nm panels?

Further, it was unclear how the cosedimentation results were interpreted. Do the authors think that POK2 fragments aggregate so much (more than dimers or tetramers) as might be expected based on the substantial pelleting without microtubules in the left side of Panel A (page 20) that the apparent microtubule interaction on the left hand side is not valid? If so, how would rampant POK2 fragment aggregation in solution alter other experiments and conclusions in this manuscript?

Was the title changed? It did not appear to be updated.

Line 48, no reference is provided for TANGLED recruitment by the PPB.

Line 48: IQD 6,7,8 localization is more broad than the CDZ, this sentence should be altered to reflect that.

Line 401-403, including Mills et al 2022 in the first set of references is warranted since it is the only one that shows that POK1 localization depends on TAN. Further, disrupting interaction between TAN and POK1 substantially reduces POK1 accumulation at the CDZ, the second sentence should be altered to reflect that. Additionally, Buschmann et al 2006 AIR9 should be discussed and referenced here.

Line 373: "While this model needs to be thoroughly tested in vivo, we note that an independent in planta study recently confirmed the presence of a second microtubule and a lipid binding domain, confirming that both traits are crucial for CDZ targeting and maintenance of POK2, thereby largely supporting our model (Livanos et al., 2025)." This sentence needs to be significantly altered. The experiments in Livanos 2025 included in vitro lipid and microtubule binding data that needs to be more thoroughly discussed here and throughout the discussion, particularly in relation to similar results. I am not opposed to the publication of these largely similar results. But to refer to this published work as an 'in planta study' with no mention of the biochemical experiments is exceptionally misleading and shortchanges both sets of results.

Second revision

Author response to reviewers' comments

Reviewer 1:

This is the revised version of a manuscript I previously reviewed. I thank the authors for their thorough responses and the additional experimental data, which convincingly support the main conclusions. Overall, the manuscript has significantly improved and - in my opinion - is suitable for publication in JCS. The study is also particularly timely, as it aligns with recent findings published in Nature Communications (Livanos et al, 2025), highlighting its relevance to current developments in the field. In my view, the work should not be delayed further.

Reviewer 2:

Having looked at the revised version of the manuscript and read the extensive response to the reviewers, I feel the authors have done a thorough job of appropriately addressing the points I raised. In my opinion, the manuscript is now appropriate for acceptance.

Reviewer 3:**SUMMARY OF THE ADVANCE MADE IN THIS PAPER AND ITS POTENTIAL SIGNIFICANCE TO THE FIELD**

Revision

SUGGESTIONS TO AUTHORS

[1] Figure 2A (middle panel) tubulin only channel looks very different from the other tubulin only channels in other parts of the figure and the rest of the manuscript. The authors are encouraged to find other images (?). This was possibly mentioned in the last review.

We have now replaced the images in Figure 2A, construct eGFP-POK2²⁵¹⁰⁻²⁷⁷¹ with representative images from another replicate of this construct's binding assay, where the tubulin signal is better visible.

[2] Similarly, in Figure 5, with the same construct (middle panel), why is the background so high for 20 nm protein?

First, contrast levels in Figure 5 and Figure 2 have been adjusted differently as the used eGFP-POK2²⁵¹⁰⁻²⁷⁷¹ concentration and intensity of bound molecules differ by at least a factor 25 between these figures. Elevated intensity levels in Figure 5 mainly show unspecific, i.e., background binding next to microtubules of this construct, that cannot be seen with the large contrast range of Figure 2.

Second, construct eGFP-POK2²⁵¹⁰⁻²⁷⁷¹ consistently showed a stronger background binding compared to the other constructs. This intrinsic trait we cannot change. To account for different background binding, we corrected for this effect in our intensity measurements as described in the last paragraph of the section "*In vitro* microtubule binding assay" in the Materials & Methods.

To reduce background binding, our flow cells are thoroughly cleaned, surface-modified, and blocked. Since variations exist across the glass coverslip and between batches, local differences in background binding cannot be prevented. Moreover, our assay is sensitive to see single molecules, i.e., small amounts of non-specifically bound molecules are visible. Thus, we compensated these effects by local background correction as pointed out above.

We now point out in the caption of Fig. 2 that brightness values of the tubulin channel have been adapted for visibility and in the caption of Fig. 5 that POK2²⁵¹⁰⁻²⁷⁷¹ is more sticky.

[3] The addition of subtilisin treatment to remove E-hooks was interesting. Was the spelling correct in the figures and manuscript?

We have corrected the typo in the figures.

[4] Thank you for providing microtubule co-sedimentation experiments. The widely variable tubulin concentrations in the bottom panels were unexpected. Why is there so little tubulin in the pellet fraction in 100 nm panels? Further, it was unclear how the cosedimentation results were interpreted.

In the previous response letter, we performed two microtubule co-sedimentation experiments that were either done by pelleting stabilised microtubules (A) directly or (B) through a 60% glycerol cushion. Both approaches unfortunately come with limitations that precluded us from reliably interpreting the data. The actual degree of POK2/tubulin co-sedimentation was unclear. Thus, we did not consider this data further in our manuscript.

Without a cushion, it is technically difficult (to impossible) to fully separate supernatant and pellet fraction. Pipetting off the supernatant does not remove all supernatant from the pellet and might perturb the pellet, mixing parts of it back into the supernatant. As a result, the outcomes of these type of experiments show some variation even across the respective samples of the same experiment. In (A), 2 μM of (hard-to-depolymerise) double-stabilised microtubules have been used as input in all conditions. We expected the same fraction of sedimented tubulin across all conditions. However, only half of the tubulin seems to be pelleted in form of microtubules at 25 nM POK2, all at 50 nM POK2 and almost none at 100 nM;

Hence, this experiment cannot be interpreted reliably due to rather arbitrary technical/experimental variations - neither in terms of tubulin sedimentation nor POK2 co-sedimentation.

These limitations may be overcome using a cushion. With a cushion, we could sufficiently separate pellet and supernatant fractions yielding reliable and reproducible tubulin sedimentation results in (B):

Concomitantly, POK2 fragments co-sedimented with double-stabilised microtubules in a concentration dependent manner.

However, analysis of the cushion after spin, revealed that the majority of POK2 fragments bound to microtubules dissociated upon passage through the cushion. Hence, the amount of POK2 fragments in the pellet shown in (B) does not reflect the amount of protein that binds microtubules at the indicated concentration in solution:

Thus, also this assay cannot be used to generate an alternative, quantitative microtubule binding affinity curve of this POK2 fragment as previously suggested by the reviewer.

The co-sedimentation results are consistent with the dynamic nature of the POK2 C-terminal domain's interaction with microtubules (Fig. 3). We found that this domain diffuses on microtubules and dissociated after seconds to tens of seconds. This duration is much shorter than the several-minute-long spin-down duration and explains the reduced amount of co-sedimented POK2²⁰⁸³⁻²⁷⁷¹ and its presence in the cushion. Our results are also consistent with the weak interaction reported via co-sedimentation of Livanos *et al.* 2025. As the fluorescence-based binding assay is much more sensitive and direct compared to the co-sedimentation assay, we preferred to focus on the fluorescence assay.

[5] Do the authors think that POK2 fragments aggregate so much (more than dimers or tetramers) as might be expected based on the substantial pelleting without microtubules in the left side of Panel A (page 20) that the apparent microtubule interaction on the left hand side is not valid? If so, how would rampant POK2 fragment aggregation in solution alter other experiments and conclusions in this manuscript?

As pointed out in our previous response to the reviewers (see point 9), we do observe higher oligomeric species or clusters in our TIRF assays *in vitro* that form *reversibly* in a microtubule- and concentration-dependent manner. Hence, we indeed expected some proportion of the POK2 fragments to pellet by themselves in the sedimentation assay without cushion. Background sedimentation activity of POK2 fragments does not automatically invalidate the outcome of our co-sedimentation assay in the presence of microtubules as discussed below. Furthermore, using a Western blot detection, we cannot distinguish between pellet signal caused by background sedimentation or by co-sedimentation activity. Hence, this assay is not suitable to quantify the microtubule affinity of POK2 fragments.

The reviewer is right that a 'substantial' amount of POK2 already sedimented in the pre-clearance step:

As already discussed in Point [4] above, it is difficult to separate pellet and supernatant without a cushion, leading to inconsistent, non-reproducible results. In this case, all protein was precleared in the absence of tubulin, so that we would expect the first and fourth lane

(both 50nM POK2) to roughly yield the same signal. This is clearly not the case, putting the quantitative value of this data into question. Apart from this fundamental issue with the assay, we also note that the pelleting of protein removes it from the oligomer equilibrium in solution, leading to a forced accumulation of sedimented POK2 over time, that would not occur in the TIRF-assays.

We disagree with the reviewer that there is a ‘rampant’ POK2 aggregation taking place in the sedimentation assays, given that after pre-clearance no further substantial amount of POK2 precipitates in the non-microtubule control:

Consistent with this observation, no larger aggregates can be seen passing through the cushion in the second co-sedimentation assay:

In our TIRF assays, larger oligomer formation was concentration dependent and, importantly, reversible. Thus, we do not think that (irreversible) aggregate formation affected our experiments or conclusions.

[6] Was the title changed? It did not appear to be updated.

Yes, it was. In response to the reviewer that our previous title was an overstatement, we had changed “connectivity” to “interaction” in the title. By “connectivity”, we originally wanted to convey our hypothesis for the *in vivo* function of the POK2 tail to network microtubules, lipids, MAP65s, and perhaps other division site players at the cortical division zone. Given that we measured interactions *in vitro*, we chose the word “interaction” to better reflect our results and highlight this relationship between the molecules we investigated, without overstating their importance. We think that this title reflects our results and is not an overstatement.

[7] Line 48, no reference is provided for TANGLED recruitment by the PPB.

We have added the respective citations (Smith *et al.*, 2001 and Walker *et al.*, 2007) in the revised version of our manuscript.

[8] Line 48: IQD 6,7,8 localization is more broad than the CDZ, this sentence should be altered to reflect that.

In the introduction, we list proteins that associate with the PPB and remain at the cortex introducing the concept of molecular markers for the future division site. To keep the introduction focussed, we mention more details, for example, the additional pools of POKs at the phragmoplast midzone, the broader distribution of IQDs, or the cell-cycle dependent sharpening of TANGELED or POK rings in the discussion, when these points become relevant. In case of the IQDs, we mention their broader distribution in line 414. Nevertheless, we now also point out in the introduction that the localisation width of the marker protein varies.

[9] Line 401-403, including Mills et al 2022 in the first set of references is warranted since it is the only one that shows that POK1 localization depends on TAN. Further, disrupting interaction between TAN and POK1 substantially reduces POK1 accumulation at the CDZ, the second sentence should be altered to reflect that. Additionally, Buschmann et al 2006 AIR9 should be discussed and referenced here.

We rewrote this section starting with line 415, to describe the contribution of TAN and AIR9 to POK1 retention at the CDZ more accurately - now including the references Mills et al., 2022, Buschmann et al., 2006 as well as Buschmann et al., 2015, which confirms earlier AIR9 localisation results in *Arabidopsis*.

[10] Line 373: "While this model needs to be thoroughly tested in vivo, we note that an independent in planta study recently confirmed the presence of a second microtubule and a lipid binding domain, confirming that both traits are crucial for CDZ targeting and maintenance of POK2, thereby largely supporting our model (Livanos et al., 2025)." This sentence needs to be significantly altered. The experiments in Livanos 2025 included in vitro lipid and microtubule binding data that needs to be more thoroughly discussed here and throughout the discussion, particularly in relation to similar results. I am not opposed to the publication of these largely similar results. But to refer to this published work as an 'in planta study' with no mention of the biochemical experiments is exceptionally misleading and shortchanges both sets of results.

We now expanded the first section of our discussion, putting our results into context of the biochemistry results presented in Livanos et al. 2025 - see lines 328 ff. We further refer to their *in vivo* data throughout the discussion, where appropriate (see line 357 f, 386 ff and 407 ff.)

Third decision letter

MS ID#: jcs.263785R2

MS Title: The plant kinesin-12 POK2 tail is a versatile interaction hub

Authors: Shu Yao Leong; Laura Muras; Benedikt Sebastian Jakob Fischer; Sehee Jang; Anastasia Gurskaya; Mayank Chugh; Serapion Pyrpapopoulos; Hauke Drechsler; Erik Schäffer
Article Type: Research Article

Dear Dr Schäffer,

I am delighted to tell you that your manuscript has been accepted for publication in Journal of Cell Science, pending standard publication integrity checks.